# FASTER APPROXIMATION OF PROBABILISTIC AND DISTRIBUTIONAL VALUES VIA LEAST SQUARES

**Weida Li**
vidaslee@gmail.com

**Yaoliang Yu**
School of Computer Science
University of Waterloo
Vector Institute
yaoliang.yu@uwaterloo.ca

## ABSTRACT

The family of probabilistic values, axiomatically-grounded in cooperative game theory, has recently received much attention in data valuation. However, it is often computationally expensive to compute exactly (exponential w.r.t. the number of data to valuate denoted by $n$). The existing generic estimator costs $O(n^2 \log n)$ utility evaluations to achieve an $(\epsilon, \delta)$-approximation under the 2-norm, while faster estimators have been developed recently for special cases (e.g., empirically for the Shapley value and theoretically for the Banzhaf value). In this work, starting from the discovered connection between probabilistic values and least square regressions, we propose a **G**eneric **E**stimator based on **L**east **S**quares (GELS) along with its variants that cost $O(n \log n)$ utility evaluations for many probabilistic values, largely extending the scope of this currently best complexity bound. Moreover, we show that each distributional value, proposed by Ghorbani et al. (2020) to alleviate the inconsistency of probabilistic values induced by using distinct databases, can also be cast as optimizing a similar least square regression. This observation leads to a theoretically-grounded framework TrELS (**Tr**aining **E**stimators based on **L**east **S**quares) that can train estimators towards the specified distributional values without requiring any supervised signals. Particularly, the trained estimators are capable of predicting the corresponding distributional values for unseen data, largely saving the budgets required for running Monte-Carlo methods otherwise. Our experiments verify the faster convergence of GELS, and demonstrate the effectiveness of TrELS in learning distributional values. Our code is available at https://github.com/watml/fastpvalue.

## 1 INTRODUCTION

In cooperative game theory, the family of probabilistic values, to which the Shapley value (Shapley 1953) and the Banzhaf value (Banzhaf 1965) belong, is uniquely characterized by the axioms of linearity, dummy, monotonicity and symmetry (Weber 1977, Theorems 5 and 10). The induced formula of probabilistic values is often deemed essential for data valuation methods (Kwon and Zou 2022a; Lin et al. 2022; Wang and Jia 2023), and is also proved effective in feature attribution (Jethani et al. 2022; Kwon and Zou 2022b; Lundberg and Lee 2017). Specifically, data valuation aims to impute an importance value to each data point $z$ in the training dataset $D_{tr}$ that represents its contribution to the performance of a model trained on $D_{tr}$, and it is empirically expected that models retrained without the "less valuable" data (e.g., data that are assigned with importance values lower than a specified threshold) in $D_{tr}$ may achieve performance gains (Ghorbani and Zou 2019).

Throughout, we identify the dataset $D_{tr}$ with $[n] = \{1, 2, \ldots, n\}$, where $n = |D_{tr}|$ is its size. Without ambiguity, for each subset $S$, its lower-case $s$ is used to denote its cardinality $|S|$, and we write $S \cup i$ and $S \backslash i$ instead of $S \cup \{i\}$ and $S \backslash \{i\}$, respectively. Each probabilistic value can be parameterized by a list of non-negative vectors $\mathcal{P} = \{\mathbf{p}^n \in \mathbb{R}_+^n\}_{n \geq 1}$ such that $\sum_{s=1}^n \binom{n-1}{s-1} p_s^n = 1$ for every $n \geq 1$, and the importance value assigned to the $i$-th data point is computed by

$$\phi_i(U^n) = \phi_i(U^n; \mathcal{P}) = \sum_{S \subseteq [n] \backslash i} p_{s+1}^n (U^n(S \cup i) - U^n(S)) \tag{1}$$

where $U^n : 2^{[n]} \to \mathbb{R}$ is a user-specified utility function and the superscript $n$ is to indicate its domain $2^{[n]}$. Typically, $U^n(S)$ outputs the performance of a chosen model trained on $S \subseteq [n]$. In this work, $U^n(\emptyset)$ denotes the performance of initialized models. Take classification tasks as an example, $U^n(S)$ could be the accuracy or the cross-entropy loss reported on a held-out dataset $D_{perf}$. It is obvious that computing $\phi(U^n)$ exactly requires $2^n$ times of evaluating the utility function $U^n$, and hence is intractable. Therefore, there has been much research devoted to developing efficient estimators. To our best knowledge, there is only one family of *generic* estimators designed for all probabilistic values: sampling lift and its weighted variant (Kwon and Zou 2022a). Specifically, the sampling lift estimator requires $O(\frac{n^2}{\epsilon^2} \log \frac{n}{\delta})$ utility evaluations to achieve an $(\epsilon, \delta)$-approximation (see Proposition 5 in the Appendix). We also note that Zhou et al. (2023, Lemma 1 and Proposition 1) analyzed convergence using $(\epsilon_1, \epsilon_2, \delta)$-approximation instead where $\epsilon_1$ and $\epsilon_2$ account for the multiplicative and additive error, respectively.

So far, many faster estimators have been proposed for specific probabilistic values. For instance, the advent of faster estimators designed specifically for the Shapley value was witnessed (Covert and Lee 2021; Kolpaczki et al. 2023; Lundberg and Lee 2017; Zhang et al. 2023b); Wang and Jia (2023, Theorem 4.9) proved that the estimator based on their proposed maximum sample reuse (MSR) principle only requires $O(\frac{n}{\epsilon^2} \log \frac{n}{\delta})$ utility evaluations for the Banzhaf value, but they also demonstrated in Appendix C.2 therein that the MSR estimator does not extend to many other probabilistic values, e.g., the family of Beta Shapley values (Kwon and Zou 2022a), in which the Shapley value resides. We also notice that Lin et al. (2022) discovered a framework of unconstrained least square regressions that leads to an estimator for a subfamily of probabilistic values (which, however, does not include the Shapley value). All in all, it is still an open question on how to efficiently approximate other probabilistic values, e.g., the Beta Shapley values.

A potential drawback of the probabilistic values is that they depend on the underlying database $D_{tr}$. Using another database, the recalculated importance values could be very inconsistent with the previous ones. To overcome this issue, Ghorbani et al. (2020) proposed the framework of distributional values, in which the importance value of any (unseen) data point $z$ is

$$\psi(z; \mathcal{D}, \mathbf{w}, U) = \mathop{\mathbb{E}}_{s \overset{\mathbf{w}}{\sim} [m]} \mathop{\mathbb{E}}_{S \sim \mathcal{D}^{s-1}} [U(S \cup z) - U(S)] \tag{2}$$

where $\mathcal{D}$ is a data distribution, $\mathbf{w} \in \mathbb{R}^m$ is a probability vector and the domain of the utility function $U$ is $\bigcup_{n \geq 1} \{S \sim \mathcal{D}^n\}$. Empirically, $\mathcal{D}$ is replaced by $\frac{1}{|B|} \sum_{z \in B} \delta_z$ where $B$ is a (large) dataset sampled from $\mathcal{D}$ (Ghorbani et al. 2020). Clearly, the computational cost for the distributional values is a big hurdle for its practical deployment.

In this paper, starting from the discovered connection between probabilistic values and least squares (see Proposition 2), we develop a **G**eneric **E**stimator based on **L**east **S**quares (GELS) along with its variants: GELS-R and GELS-Shapley. Precisely, GELS-R is to approximate the ranking of probabilistic values, while GELS-Shapley is specific to the Shapley value. In addition, we demonstrate that each distributional value can also be cast into a similar least square regression, which serves as the theoretical ground for formulating the framework TrELS (**Tr**aining **E**stimators based **L**east **S**quares) that can train estimators towards the specified distributional value without requiring any supervised signals. In other words, unlike Monte-Carlo methods, TrELS allows the use of trained estimators for prediction. Our main contributions are summarized as follows:

1. We propose GELS and GELS-R for all probabilistic values, and prove that both of them require $O(\frac{n}{\epsilon^2} \log \frac{n}{\delta})$ utility evaluations to achieve an $(\epsilon, \delta)$-approximation for many probabilistic values, matching the currently best bound for special cases; see Algorithms 1 and 2. In addition, we also develop GELS-Shapley in Algorithm 3 specific to the Shapely value.

2. By casting the distributional values into optimizing least square regressions, we design TrELS for training estimators towards distributional values without supervision. See Algorithm 4 and Theorem 1.

3. Our experiments verify the faster convergence of GELS and GELS-R, and we also show that the distributional values can be well-learned using TrELS.

4. As a minor side note, we also extend the approximation-without-requiring-marginal (ARM) estimator, designed specifically for the Shapley value (Kolpaczki et al. 2023), to the whole family of probabilistic values. See Proposition 8 in the Appendix.

## 2  BACKGROUND

The sampling lift estimator (Moehle et al. 2022) refers to any approximation algorithm designed according to

$$\phi_i(U^n) = \sum_{S \subseteq [n] \setminus i} p_{s+1}^n \left( U^n(S \cup i) - U^n(S) \right) = \mathbb{E}_{S \subseteq [n] \setminus i}[U^n(S \cup i) - U^n(S)].$$

Its weighted variant is to use $n\binom{n-1}{s}p_{s+1}^n(U^n(S \cup i) - U^n(S))$ with $P(S) = \frac{1}{n}\binom{n-1}{s}^{-1}$ instead (Kwon and Zou 2022a). Particularly, they are the same for the Shapley value. As summarized in Proposition 5, the sampling lift requires $O(\frac{n^2}{\epsilon^2} \log \frac{n}{\delta})$ utility evaluations to achieve an $(\epsilon, \delta)$-approximation. Recently, Kolpaczki et al. (2023) proposed an ARM estimator specifically for the Shapley value, but we notice that it can be easily generalized for all probabilistic values. Therefore, we present the *generic* formula of ARM in the main paper while leaving its justification to Proposition 8 in the Appendix. The ARM estimator is to approximate by using

$$\phi_i(U^n) = \mathbb{E}_{S \sim P_{ARM}^+ | i \in S}[U^n(S)] - \mathbb{E}_{S \sim P_{ARM}^- | i \notin S}[U^n(S)]$$

where $P_{ARM}^+(S) \propto p_s^n$ for all *non-empty* subsets $S \subseteq [n]$ and $P_{ARM}^-(S) \propto p_{s+1}^n$ for all $S \subsetneq [n]$.

In addition, Lin et al. (2022) developed the average marginal effect (AME) that casts a *subfamily* of probabilistic values as the uniquely optimal solution to

$$\arg\min_{\mathbf{v} \in \mathbb{R}^n} \mathbb{E}[(Y - \mathbf{X}^\top \mathbf{v})^2],$$

where $\mathbf{X}$ and $Y$ are random variables whose distribution is induced by i) sampling $t \sim P$ where $P$ is any probability distribution on the open interval $(0, 1)$, ii) sampling a subset $S \subseteq [n]$ by including each data point in $[n]$ with probability $t$, and iii) setting $Y = U^n(S)$ and $X_i = \frac{1}{t \cdot M_P}$ if $i \in S$ and $\frac{-1}{(1-t)M_P}$ otherwise where $M_P = \mathbb{E}_{t \sim P}[\frac{1}{t(1-t)}]$. Particularly, the uniquely optimal solution is just a probabilistic value parameterized by $p_s^n = \int_0^1 t^{s-1}(1-t)^{n-s} dP(t)$. If $P$ is uniform, the corresponding $\mathbf{p}^n$ is the one defining the Shapley value, but $M_P = \infty$, and thus AME does not work for the Shapley value.

On the other hand, Wang and Jia (2023) proposed the maximum sample reuse (MSR) principle which aims to find a distribution $P_{MSR}$ on $2^{[n]}$ such that for every $i \in [n]$,

$$P_{MSR}(S \mid i \in S) = p_s^n \text{ and } P_{MSR}(S \mid i \notin S) = p_{s+1}^n,$$

with which Eq. (1) can be rewritten as $\phi_i(U^n) = \mathbb{E}_{S|i \in S}[U^n(S)] - \mathbb{E}_{S|i \notin S}[U^n(S)]$. Specifically, $P_{MSR}$ is uniform over $2^{[n]}$ for the Banzhaf value, and Wang and Jia (2023, Theorem 4.9) showed that the resulting estimator only requires $O(\frac{n}{\epsilon^2} \log \frac{n}{\delta})$ utility evaluations. However, they also demonstrated in Appendix C.2 therein that such $P_{MSR}$ exists if and only if $p_{s+1}^n = \eta(n) \cdot p_s^n$ for every $1 \le s \le n-1$ where $\eta(n) \in \mathbb{R}$, which excludes all Beta Shapley values.

A recently-proposed faster algorithm specific to the Shapley value is the complement estimator (Zhang et al. 2023b) using the complement formula

$$\phi_i^{Sh}(U^n) = \sum_{S \subseteq [n]: i \in S} \frac{(s-1)!(n-s)!}{n!} \left( U^n(S) - U^n([n] \setminus S) \right),$$

which was discovered half a century ago by Harsanyi (1963, Eq. (4.1)). Earlier, Lundberg and Lee (2017) proposed the kernelSHAP estimator that exploits the fact that the Shapley value $\phi^{Sh}(U^n)$ is the uniquely optimal solution to

$$\arg\min_{\mathbf{v} \in \mathbb{R}^n} \sum_{\emptyset \subsetneq S \subsetneq [n]} \binom{n-2}{s-1}^{-1} \left( U^n(S) - U^n(\emptyset) - \sum_{i \in S} v_i \right)^2 \tag{3}$$

$$\text{s.t. } \sum_{i \in [n]} v_i = U^n([n]) - U^n(\emptyset),$$

which was first discovered by Charnes et al. (1988, Theorem 4). Since it is difficult to analyze whether the kernelSHAP is unbiased or not, Covert and Lee (2021, Eq. (9)) later proposed a (provably) unbiased variant, and suggested that the paired sampling technique can enhance both. Very recently, Fumagalli et al. (2023, Theorem 4.5) and Zhang et al. (2023a, Eqs. (11) and (12)) simplified the formula of the unbiased kernelSHAP.

## 3 MAIN RESULTS

In this section, we first show how to efficiently estimate the ranking induced by any probabilistic value. Then, by introducing a null data point we show how to turn our ranking estimator into a *bona fide* estimator, while retaining the same efficiency for many probabilistic values. Lastly, we extend our theory to an unsupervised framework that trains estimators towards distributional values.

### 3.1 WHEN RANKING SUFFICES

Practitioners often need to screen training data before feeding them to a model, to remove outliers, low-quality data, or even adversarial examples that are deemed harmful to training. Data valuation is a natural way to achieve this goal, i.e., only data assigned with high importance values could be considered potentially "valuable." If one has a good estimate of the proportion of "valuable" data, then the relative ranking, instead of the more precise and demanding importance values, suffices. Our first results below pave the way to efficiently estimate the relative ranking underlying any probabilistic value. Particularly, $\mathcal{G}^n = \{U^n : 2^{[n]} \to \mathbb{R}\}$ is the set that contains all possible utility functions provided there are $n$ data to valuate.

**Proposition 1.** *Define, for every $n \geq 1$, $U^n \in \mathcal{G}^n$ and $i \in [n]$,*

$$\mathcal{R}_i(U^n) = \mathcal{R}_i(U^n; \mathcal{P}) = \sum_{\emptyset \subsetneq S \subsetneq [n]} m_s^n \cdot \mathbb{1}_{i \in S} \cdot U^n(S) \tag{4}$$

*where $m_s^n = m_s^n(\mathcal{P}) = p_s^n + p_{s+1}^n$ for every $s \in [n-1]$. Then, for every $n \geq 1$ and $U^n \in \mathcal{G}^n$, $\mathcal{R}(U^n)$ and $\phi(U^n)$ produce the same ranking. Precisely, $\mathcal{R}(U^n) = \phi(U^n) + g(U^n)\mathbf{1}_n$ where $g(U^n) = g(U^n; \mathcal{P}) \in \mathbb{R}$. As a side note, it holds for any $\mathbf{p}^n \in \mathbb{R}^n$.*

We emphasize that Proposition 1 is closely related to Proposition 2 below in the sense that they immediately imply each other; we *first* noticed a more general version of proposition 2 that applies to all least square values (Ruiz et al. 1998, Definition 5), a broader family that includes all the additive-efficient-normalized probabilistic values; see Appendix D for more details.

**Proposition 2.** *Consider the problem*

$$\underset{\mathbf{v} \in \mathbb{R}^n}{\arg\min} \sum_{\emptyset \subsetneq S \subsetneq [n]} m_s^n \cdot \left(U^n(S) - \sum_{i \in S} v_i\right)^2, \tag{5}$$

*its uniquely optimal solution shares the same ranking as $\phi(U^n)$. As a side note, it holds true for any $\mathbf{p}^n \in \mathbb{R}^n$ that produces a non-negative weight vector $\mathbf{m}^n$ with $\sum_{s=1}^{n-1} m_s^n > 0$.*

Algorithm 1 summarizes the estimator induced by Propositions 1 and 2. Note that for GELS-R each utility evaluation $U^n(S)$ can be used for updating $s$ estimate of the specified probabilistic value. By contrast, the (weighted) sampling lift estimator spends two utility evaluations to update the estimate of only one data point.

### 3.2 WHEN PROBABILISTIC VALUES ARE DESIRED

**Remark 1.** *To recover $\phi(U^n)$ from $\mathcal{R}(U^n)$, we introduce a null data point labeled as $n+1$ to extend each $U^n$ into $\overline{U}^{n+1}$ such that $\overline{U}^{n+1}(S) = U^n(S \cap [n])$ for every $S \subseteq [n+1]$. Meanwhile, we construct $\mathbf{p}^{n+1} \in \mathbb{R}^{n+1}$ such that $p_s^n = p_s^{n+1} + p_{s+1}^{n+1}$ for every $s \in [n]$, and note that $\mathbf{p}^{n+1}$ may contain negative weights. Proposition 1 demonstrates that $\mathcal{R}(\overline{U}^{n+1}) = \phi(\overline{U}^{n+1}) + g(\overline{U}^{n+1})\mathbf{1}_{n+1}$. Particularly, the definition of Eq. (1) makes that $\phi_{n+1}(\overline{U}^{n+1}) = 0$ and the structure $p_s^n = p_s^{n+1} + p_{s+1}^{n+1}$ leads to $\phi_i(U^n) = \phi_i(\overline{U}^{n+1})$ for every $i \in [n]$. Therefore, $g(\overline{U}^{n+1}) = \mathcal{R}_{n+1}(\overline{U}^{n+1})$, and we have the recovery formula $\phi_i(U^n) = \phi_i(\overline{U}^{n+1}) = \mathcal{R}_i(\overline{U}^{n+1}) - \mathcal{R}_{n+1}(\overline{U}^{n+1})$. This idea is summarized in Proposition 3 and Algorithm 2.*

**Proposition 3.** *The uniquely optimal solution $\mathbf{v}^*$ to the problem*

$$\underset{\mathbf{v} \in \mathbb{R}^{n+1}}{\arg\min} \sum_{\emptyset \subsetneq S \subsetneq [n+1]} p_s^n \cdot \left(U^n(S \cap [n]) - \sum_{i \in S} v_i\right)^2 \tag{6}$$

*satisfies that $\phi_i(U^n) = v_i^* - v_{n+1}^*$ for every $i \in [n]$.*

---

**Algorithm 1:** GELS-R (**G**eneric **E**stimator based on **L**east **S**quares for **R**ankings)

---

**Input:** A dataset $D_{tr} \equiv [n]$ to be valued, a utility function $U^n \in \mathcal{G}^n$, a weight vector
$\quad$ $\mathbf{q} \in \mathbb{R}^{n-1}$ defined by $q_s = \binom{n}{s}(p_s^n + p_{s+1}^n)$, and a total number $T$ of samples

**Output:** An unbiased estimate $\hat{\mathbf{r}}$ to $\mathcal{R}(U^n)$ up to some scalar

1 Normalize $\mathbf{q}$ into a probability vector $\mathbf{q} \leftarrow \mathbf{q}/\sum_{s=1}^{n-1} q_s$

2 $\hat{\mathbf{r}} \leftarrow \mathbf{0}_n, \mathbf{t} \leftarrow \mathbf{0}_n$

3 **for** $k = 1, 2, \ldots, T$ **do**

4 $\quad$ Sample $s_k \in [n-1]$ using the probability vector $\mathbf{q}$

5 $\quad$ Uniformly sample $S_k$ from $\{R \subseteq [n] \mid |R| = s_k\}$

6 $\quad$ **for** $i \in S_k$ **do**

7 $\quad\quad$ $t_i \leftarrow t_i + 1$ and $\hat{r}_i \leftarrow (1 - \frac{1}{t_i})\hat{r}_i + \frac{1}{t_i}U^n(S_k)$

---

**Algorithm 2:** GELS (**G**eneric **E**stimator based on **L**east **S**quares)

---

**Input:** A $D_{tr} \equiv [N]$ to be valued, a utility function $U^n \in \mathcal{G}^n$, a weight vector $\mathbf{q} \in \mathbb{R}^n$
$\quad$ defined by $q_s = \binom{n+1}{s}p_s^n$, and a total number $T$ of samples

**Output:** An unbiased estimate $\hat{\phi}$ to $\phi(U^n)$

1 Introduce a null datum labeled by $n+1$, and extend $U^n$ to $\overline{U}^{n+1}$ $\quad$ // $\overline{U}^{n+1}(S) = U^n(S \cap [n])$

2 Obtain $\hat{\mathbf{r}} \in \mathbb{R}^{n+1}$ from Algorithm 1 using $[n+1], \overline{U}^{n+1}, \mathbf{q}$ and $T$

3 $\hat{\phi}_i \leftarrow (\sum_{s=1}^n \frac{s}{n+1}q_s)(\hat{r}_i - \hat{r}_{n+1})$ for each $i \in [n]$ $\quad\quad$ // using unnormalized $\mathbf{q}$

---

**Remark 2.** *For the Shapley value, since $\sum_{i \in [n]} \phi_i^{Sh}(U^n) = U^n([n]) - U^n(\emptyset)$ and by Proposition 1 $\phi(U^n) = \mathcal{R}(U^n) + g(U^n)\mathbf{1}_n$, we have $U^n([n]) - U^n(\emptyset) = \sum_{i \in [n]} \mathcal{R}_i(U^n) + g(U^n) \cdot n$, and thus $g(U^n) = \frac{1}{n}(U^n([n]) - U^n(\emptyset) - \sum_{i \in [n]} \mathcal{R}_i(U^n))$. Therefore, we can recover $\phi^{Sh}(U^n)$ without introducing a null data point, which is summarized in Algorithm 3.*

**Proposition 4.** *Assume that $\|U^n\|_\infty \leq u$ for every $U^n \in \bigcup_{n \geq 1} \mathcal{G}^n$. We have the following results: i) GELS requires $O(\frac{\tau(n)n}{\epsilon^2} \log \frac{n}{\delta})$ utility evaluations to achieve an $(\epsilon, \delta)$-approximation, i.e., $P(\|\hat{\phi}(U^n) - \phi(U^n)\|_2 \geq \epsilon) \leq \delta$; ii) for GELS-R estimator, it requires $O(\frac{\kappa(n)n}{\epsilon^2} \log \frac{n}{\delta})$ utility evaluations instead; iii) plus, the corresponding convergence of GELS-Shapley is $O(\frac{n}{\epsilon^2} \log(n)^2 \log \frac{n}{\delta})$.*

**Remark 3.** *Appendix H presents that both $\tau(n)$ and $\kappa(n)$ are proportional to the inverse square of the average reuse rate of utility evaluations. In other words, the more estimates on average in $\hat{\phi}$ each utility evaluation are used to update, the more efficient GELS and GELS-R will be. Precisely, we study the asymptotic behaviors of $\tau(n)$ and $\kappa(n)$ for semi-values, which include all probabilistic values mentioned and whose $\mathcal{P}$ can be summarized by a probability measure $\mu$ over the closed interval $[0,1]$ by $p_s^n = \int_0^1 t^{s-1}(1-t)^{n-s}d\mu(t)$. To sum, i) the Banzhaf value and the Beta Shapley values with $\alpha, \beta > 1$ corresponds to $\tau(n), \kappa(n) \in \Theta(1)$; ii) for the Beta Shapley values with $\alpha, \beta \geq 1$, it becomes $\tau(n), \kappa(n) \in O(\log(n)^2)$ instead; iii) particularly, $\kappa(n) \in \Theta(1)$ for the Shapley value (which is the Beta Shapley value with $\alpha = \beta = 1$), which indicates GELS-R requires $O(\frac{n}{\epsilon^2} \log \frac{n}{\delta})$ utility evaluations to achieve an $(\epsilon, \delta)$-approximation; however, we emphasize that the convergence of GELS-R and GELS-Shapley cannot be compared directly as the definition of $(\epsilon, \delta)$-approximation does not account for multiplicative error; iv) additionally, while using the paired sampling technique, GELS-Shapley exactly recovers unbiased KernelSHAP, and thus Proposition 4 also proves that the convergence of unbiased KernelSHAP with the paired sampling technique is $O(\frac{n}{\epsilon^2} \log(n)^2 \log \frac{n}{\delta})$. We refer the reader to Appendices F, I and J for more details.*

### 3.3 DISTRIBUTIONAL VALUE

We are now ready to combine the previous results with the notion of distributional values Eq. (2) to establish an unsupervised framework for *training* estimators towards distributional values. We refer the reader to Appendix A for more details of distributional values. Such trained estimators can valuate any unseen data point sampled from the same (or close-enough) data distribution in a single forward pass, which largely saves the budgets required for running Monte-Carlo methods otherwise.

---

**Algorithm 3:** GELS-Shapley (**G**eneric **E**stimator based on **L**east **S**quares for the **Shapley** value)

---

**Input:** A $D_{tr} \equiv [N]$ to be valuated, a utility function $U^n \in \mathcal{G}^n$, a weight vector $\mathbf{q} \in \mathbb{R}^{n-1}$
        defined by $q_s = \frac{n}{s(n-s)}$, and a total number $T$ of samples

**Output:** An unbiased estimate $\hat{\phi}$ to the Shpaley value of $U^n$

1 Obtain $\hat{\mathbf{r}} \in \mathbb{R}^n$ from Algorithm 1 using $[n]$, $U^n$, $\mathbf{q}$ and $T$

2 $\hat{r}_i \leftarrow \hat{r}_i \cdot H_{n-1}$ for each $i \in [n]$          // $H_{n-1} = \sum_{s=1}^{n-1} \frac{1}{s}$

3 $\hat{\phi}_i \leftarrow \hat{r}_i + O$ for each $i \in [n]$          // $O = (U^n([n]) - U^n(\emptyset) - \sum_{i=1}^n \hat{r}_i)/n$

---

As counterparts in feature attribution, Covert et al. (2023) and Jethani et al. (2022) exploited the least square regression (3) together with the additive efficient normalization proposed by Ruiz et al. (1998, Definition 11) to train neural networks that can then predict the Shapley value of any unseen instance in a single forward pass.

In practice, the data distribution $\mathcal{D}$ in Eq. (2) is replaced by an empirical distribution $\mathcal{B} = \frac{1}{|B|} \sum_{z \in B} \delta_z$ where $B$ is a sufficiently large dataset sampled by $\mathcal{D}$. Substituting $\mathcal{B}$ for $\mathcal{D}$ in Eq. (2), observe that there exists another probability vector $\boldsymbol{\omega} \in \mathbb{R}^m$ such that

$$\psi(z; \mathcal{B}, \mathbf{w}, U) = \varphi(z; \mathcal{B}, \boldsymbol{\omega}, U) := \mathbb{E}_{s \overset{\boldsymbol{\omega}}{\sim} [m]} \mathbb{E}_{S \overset{\mathcal{U}}{\sim} \mathcal{B}_{s-1}} [U(S \cup z) - U(S)] \tag{7}$$

where $\mathcal{U}$ represents the uniform sampling and $\mathcal{B}_{s-1} = \{R \subseteq B \mid |R| = s - 1\}$. We point out $\varphi$ is more natural to analyze. If $z \notin B$, Eq. (7) is equal to calculating some probabilistic value for $z$ with the domain of utility function being $2^{B \cup z}$; if $z \in B$, it produces some probabilistic value of $z$ up to some scalar, with the domain of utility function being $2^B$.

**Theorem 1.** *Consider the case when $B = [n]$ (recall that $D_{tr} \equiv [n]$) for the empirical distributional values $\varphi$ defined in Eq. (7). Let $\mathbf{d} \in \mathbb{R}^m$ be a probability vector that satisfies $d_s \propto \frac{n-s+1}{s} \omega_s$ for every $s \in [m]$, and $\mathbf{v}^*$ be the uniquely optimal solution to*

$$\underset{\mathbf{v} \in \mathbb{R}^{n+1}}{\arg \min} \ \mathbb{E}_{s \overset{\mathbf{d}}{\sim} [m]} \mathbb{E}_{S \overset{\mathcal{U}}{\sim} \overline{\mathcal{B}}_s} \left( U^n(S \cap [n]) - \sum_{i \in S} v_i \right)^2, \tag{8}$$

*where $\overline{\mathcal{B}}_s = \{R \subseteq [n+1] \mid |R| = s\}$. There is, for every $i \in [n]$,*

$$C \cdot (v_i^* - v_{n+1}^*) = \varphi(i; \mathcal{B}, \boldsymbol{\omega}, U^n) \ where \ C = \sum_{s=1}^m \frac{n-s+1}{n} \omega_s.$$

*Additionally, setting $p_s^n \propto \binom{n}{s-1}^{-1} \omega_s$ if $s \in [m]$ and $0$ otherwise such that $\mathbf{p}^n$ defines a probabilistic value, $\hat{\mathbf{r}}$ obtained in Algorithm 2 meets that $\mathbb{E}[\hat{r}_i] - \mathbb{E}[\hat{r}_{n+1}] = \varphi(i; \mathcal{B}, \boldsymbol{\omega}, U^n)$ for every $i \in [n]$.*

Theorem 1 constitutes a theoretically unsupervised ground for training estimators towards distributional values. Precisely, let $\phi_{\boldsymbol{\theta}}(\mathbf{x}, y)$ be a trainable model parameterized by $\boldsymbol{\theta}$; $\mathbf{x}$ and $y$ stands for features and label, respectively; we propose training the estimator $\phi_{\boldsymbol{\theta}}$ based on the specified utility function $U^n \in \mathcal{G}^n$ through optimizing

$$\underset{\boldsymbol{\theta}, \phi_o}{\arg \min} \ \mathbb{E}_{s \overset{\mathbf{d}}{\sim} [m]} \mathbb{E}_{S \overset{\mathcal{U}}{\sim} \overline{\mathcal{B}}_s} \left( U^n(S \cap [n]) - \phi_o \mathbb{1}_{n+1 \in S} - \sum_{i \in S \cap [n]} \phi_{\boldsymbol{\theta}}(\mathbf{x}_i, y_i) \right)^2. \tag{9}$$

As will be seen in the experiments, for any data point $z \notin B$, the trained $\phi_{\boldsymbol{\theta}}$ indeed predicts quite accurately the corresponding probabilistic value of $z$ with the domain of utility function being $2^{B \cup z}$!

Generally, it is impractical to train estimators using the exact distributional values, or regressing with a reasonable approximation as supervised signals, since it is extremely expensive to calculate them exactly. For example, in our experiments where $N = 10,000$ and $m = 1,000$, each unseen data point requires $2 \sum_{s=0}^{999} \binom{10,000}{s}$ utility evaluations to compute exactly. Therefore, the merits of training estimators through optimizing the problem (9) are clear: i) we do not have to be concerned with what the exact distributional values are, and ii) Theorem 1 guarantees that each estimator will be trained towards the exact distributional values. The outline of such an unsupervised framework is summarized in Algorithm 4.

---

**Algorithm 4:** TrELS (**Tr**aining **E**stimators based on **L**east **S**quares)

---

**Input:** A database $D_{tr} \equiv (\mathbf{x}_i, y_i)_{1 \leq i \leq n}$, a probability vector $\boldsymbol{\omega} \in \mathbb{R}^m$ with $m < n$, a utility
function $U^n \in \mathcal{G}^n$, a trainable model $\phi_{\boldsymbol{\theta}}$ with an additional trainable parameter $\phi_o$, a
batch size $Z$ and a total number $T$ of batches used for training.

1   Compute $\mathbf{d} \in \mathbb{R}^m$ by letting $d_s = \frac{n-s+1}{s}\omega_s$ for every $s \in [m]$

2   Normalize: $\mathbf{d} \leftarrow \mathbf{d}/\sum_{s=1}^{m} d_s$

3   **for** $t = 1, 2, \ldots, T$ **do**

4      $loss \leftarrow 0$

5      **for** $j = 1, 2, \ldots, Z$ **do**

6          Sample $s_j$ from $[m]$ according to the distribution vector $\mathbf{d}$

7          Sample $S_j$ uniformly from $\{R \subseteq [n+1] \mid |R| = s_j\}$

8          $loss \leftarrow loss + \left(U^n(S_j \cap [n]) - \phi_o \mathbb{1}_{n+1 \in S_j} - \sum_{i \in S_j \cap [n]} \phi_\theta(\mathbf{x}_i, y_i)\right)^2$

9      $loss \leftarrow loss/Z$

10      Update $\boldsymbol{\theta}$ and $\phi_o$ using $loss$

**Evaluation Phase:** $C \cdot (\phi_{\boldsymbol{\theta}}(\mathbf{x}, y) - \phi_o)$              //   $C = \sum_{s=1}^{m} \frac{n-s+1}{n}\omega_s$

---

## 4   EVALUATION

In this section we perform experiments to i) verify the faster convergence of Algorithms 1 and 2, and ii) demonstrate that distributional values can be effectively predicted by trained estimators using TrELS. The classification datasets employed are from open resources, which are iris, wind (both are from OpenML), FMNIST (Xiao et al. 2017) and MNIST. For all utility functions, their outputs are set to be the performance of the trained models reported on a dataset $D_{perf}$ disjoint from $D_{tr}$. Precisely, $U^n(S)$ reports the performance of the specified model trained on $S \subseteq D_{tr} \equiv [n]$, and we leave the specified model untrained for evaluating $U^n(\emptyset)$. Without being stated explicitly, the performance of trained models is measured by classification accuracy. In addition, we fix the random seed to be 2024 to make all utility functions *deterministic*. To be computationally efficient, we adopt one-mini-batch one-epoch learning for evaluating utility functions (Ghorbani and Zou 2019). Logistic regression is implemented for iris and wind, while LeNet (LeCun et al. 1998) is employed for FMNIST and MNIST.

### 4.1   FASTER CONVERGENCE OF THE PROPOSED ESTIMATORS

This experiment is performed on all the above-mentioned datasets. The SGD optimizer with a learning rate $1.0$ is employed for iris and wind, whereas we set the learning rate to be $0.1$ instead for MNIST and FMNIST. To evaluate the specified probabilistic values exactly, we set $|D_{tr}| = |D_{perf}| = 24$. All estimators are run 30 times with different random seeds, which range from 0 to 29. We compare to the following benchmarks:

- those designed specifically for the Shapley value, including the complement (Zhang et al. 2023b), kernelSHAP (Lundberg and Lee 2017), unbiased KernelSHAP (Covert and Lee 2021), group testing (Jia et al. 2019), and simSHAP (Zhang et al. 2023a);

- those work for all probabilistic values, including the sampling lift (see Proposition 5) and its weighted variant (Kwon and Zou 2022a), and the generalized ARM (see Proposition 8);

- AME (Lin et al. 2022), restricted to a subfamily of probabilistic values;

- MSR (Wang and Jia 2023), proposed specifically for the Banzhaf value.

The probabilistic values we consider are: i) Beta$(1, 1)$, which equals to the Shapley value, ii) Beta$(2, 2)$,[1] iii) Beta$(4, 1)$, and iv) the Banzhaf value. Particularly, AME does not apply to Beta$(4, 1)$ and the Shapley since both of them lead to $M_P = \infty$.

---

[1]This probabilistic value is special: i) it can be derived from the best min-polynomial approximation of the Lovász extension of $U^n$ (Marichal and Mathonet 2008, Corollary 19), and ii) it is uniquely characterized by a set of axioms proposed in multi-criteria decision making (Grabisch and Labreuche 2001, Theorem 2)

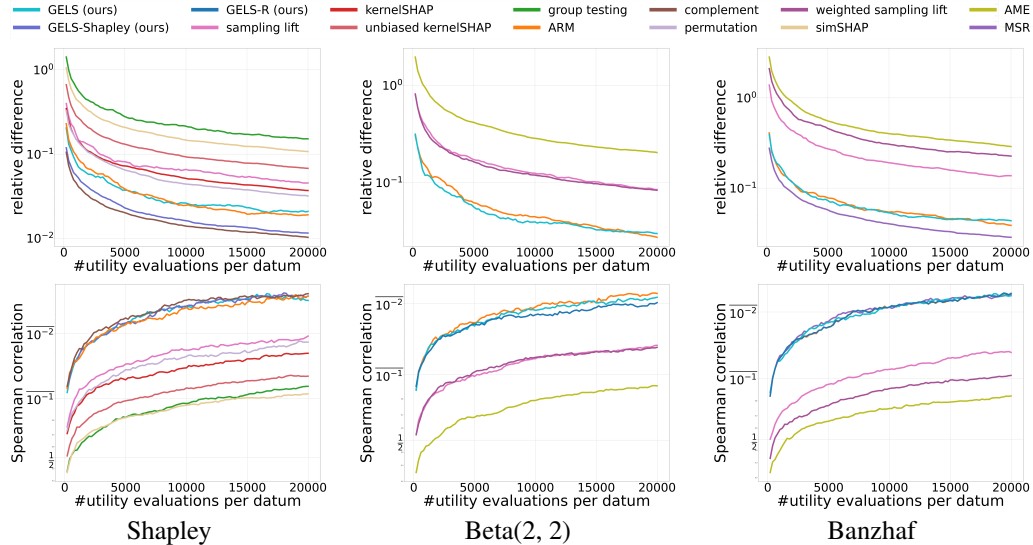

Figure 1: Comparison of different estimators using the dataset wind. The relative difference is plotted in log scale while the Spearman correlation is in logit scale. In addition, $\overline{r} = 1 - r$.

**Remark 4.** *Covert and Lee (2021) proposed the paired sampling technique to enhance (unbiased) kernelSHAP, and we point out that this technique can also be employed for our methods, (weighted) sampling lift, group testing, AME, MSR and simSHAP. We only compare the plain estimators in the main paper, and experiments including the paired sampling are provided in Appendix J.*

We present some of the results in Figure 1, while deferring others to Appendix J. These results support that GELS, GELS-R are currently among the (empirically) fastest tier in the group of *generic* estimators. For the Shapley value, we omit the results of GELS-R since GELS-R and GELS-Shapley share the same ranking; GELS-Shapley converges significantly faster than GELS; however, GELS-Shapley may converge slower compared with the complement and kernelSHPA; notice that their faster convergence comes at the cost of using $\Theta(n^2)$ memory storage instead of $\Theta(n)$; precisely, the complement requires another $O(n^2)$ time complexity to aggregate all $n^2$ estimates, whereas kernelSHAP needs an extra $O(n^3)$ time complexity for calculating the inverse of matrices. For the Banzhaf value, though GELS converges slower than MSR in terms of relative error, the performance of GELS and GELS-ranking in terms of Spearman correlation is not worse than MSR, and sometimes better. For the Beta$(2, 2)$, GELS and GELS-R achieve the best. We notice that the generalized ARM almost enjoys the same empirical convergence across different experiment settings, except that it is slower than GELS-Shapley in terms of relative error.

## 4.2 Training Estimators

In this experiment, we demonstrate the effectiveness of training estimators using Algorithm 4 on FMNIST and MNIST. The corresponding results for MNIST are included in Appendix J. For the utility functions, the SGD optimizer with a learning rate $0.01$ and one-mini-batch one-epoch learning is adopted, and we take $|D_{tr}| = 10,000$ and $|D_{perf}| = 500$ from the training set. LeNet is the architecture we employ as a trainable estimator $\phi_{\boldsymbol{\theta}}$, and the input of the softmax layer is taken as the output of $\phi_{\boldsymbol{\theta}}$. For training estimators, we employ the Adam optimizer (Kingma and Ba 2014) with a learning rate $0.001$. The batch size $Z$ is set to be $10,000$, and we randomly generate $1,000$ batches. After these batches have all been fed into the estimators, we permute and reuse them to continue training. Therefore, we have in total $1,000$ utility evaluations per data point for training estimators. Lastly, the probability vector $\boldsymbol{\omega} \in \mathbb{R}^{1,000}$ we employ is $\omega_s \propto s^{-\frac{1}{2}}$. Moreover, $200$ data, denoted by $D_{val}$, are taken from the training dataset for selecting the best trained models. For the results in the second row of Figure 2, we report on another $200$ data extracted from the test dataset, for which we refer to as $D_{test}$. To sum, $D_{tr}, D_{perf}, D_{val}$ and $D_{test}$ are all disjoint, which means $D_{val}$ and $D_{test}$ are composed of unseen data. For $D_{val}$ and $D_{test}$, we randomly run $600,000$ utility evaluations for each data point using Eq. (7) to generate estimates of the specified distributional value, which are

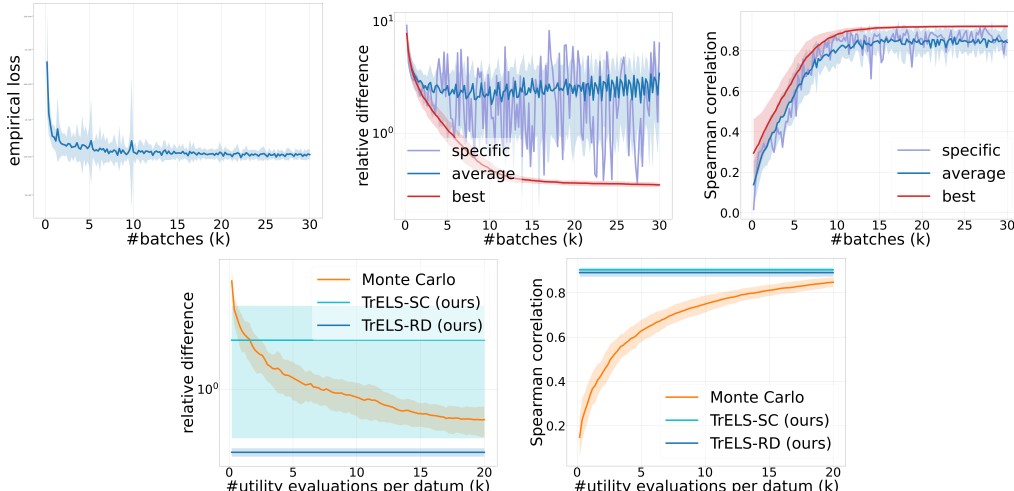

Figure 2: i) The first row: The first plot is the empirical loss of the sampled batch, while the others present the relative distance $\|\hat{\varphi} - \varphi\|_2/\|\varphi\|_2$ and the Spearman correlation between $\varphi$ and $\hat{\varphi}$. $\hat{\varphi}$ denotes the one predicted by the estimators trained on FMNIST. The first two plots are in log scale. The label "best" reports the best one achieved by the trained estimators at each time point, whereas "specific" is the one with the random seed being $0$. ii) The second row: the two plots compare the performance of the estimators trained on FMNIST with the Monte-Carlo method using Eq (2). Specifically, TrELS-SC uses trained estimators that achieve the best Spearman correlation on $D_{val}$, whereas $RD$ stands for relative difference.

regarded as the ground-truths $\varphi$. All results are reported with mean and standard deviation using 30 different random seeds ranging from 0 to 29.

The performance curves of the estimators during training are shown in Figure 2. First of all, the increasing curve in terms of the Spearman correlation indicates that the estimators trained under TrELS are able to gradually learn the exact ranking. Secondly, the decreasing curve (the best one) of the relative difference suggests that the transform $C \cdot (\phi_{\boldsymbol{\theta}}(\mathbf{x}, y) - \phi_o)$ is essential for training!

Next, we examine the efficiency of the trained estimators by comparing them with the Monte-Carlo method based on Eq. (7). Specifically, for each random seed, the trained estimators that achieves the best performance on $D_{val}$ are selected. The comparison is shown in the second row of Figure 2. As clearly shown, the Monte-Carlo methods using $20,000$ utility evaluations for each data point are still inferior to the estimators trained under our TrELS. Observe that TrELS-SC performs poorly in terms of relative difference, which is expected since the reported relative differences during training are significantly unstable.

## 5    CONCLUSION

In this work, we start from the least square regression to develop GELS with its variants for all probabilistic values. GELS-R is to approximate the ranking of probabilistic values, whereas GELS-Shapley is specifically designed for the Shapley value. The faster convergence of the proposed estimators is theoretically guaranteed, and is also verified empirically in our experiments. Besides, we also demonstrate how to cast each distributional value into a least square problem, making it the first-time theoretically-grounded to train estimators towards distributional values in an unsupervised manner, the framework of which is introduced as TrELS. Notably, our experiments show that the estimators trained under TrELS learn the specified distributional values quite well in terms of both relative difference and Spearman correlation. Our work significantly broadens the practicality of deploying value-based data valuation methods on rather large datasets.

### ACKNOWLEDGEMENTS

We thank the reviewers and the area chair for thoughtful comments that have improved our final presentation. YY gratefully acknowledges NSERC and CIFAR for funding support.

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

# Table of Contents

## A  DISTRIBUTIONAL VALUES

Originally, (Ghorbani et al. 2020, Definition 2.2) proposed the Distributional Shapley value defined by

$$\nu(z; \mathcal{D}, m, U) = \mathop{\mathbb{E}}_{S \sim \mathcal{D}^{m-1}} [\phi(z; U, S \cup z)]$$

$$\text{where } \phi(z; U, S \cup z) = \sum_{T \subseteq S} \frac{t!(m-1-t)!}{m!} (U(T \cup z) - U(T)).$$

Note that $\phi$ is supposed to be the Shapley value if $z \notin S$ and $|S| = m - 1$. Then, Ghorbani et al. (2020, Theorem 2.3) proved that

$$\nu(z; \mathcal{D}, m, U) = \mathop{\mathbb{E}}_{k \overset{\mathcal{U}}{\sim} [m]} \mathop{\mathbb{E}}_{S \sim \mathcal{D}^{k-1}} [U(S \cup z) - U(S)] \tag{10}$$

where $\mathcal{U}$ refers to the uniform distribution. To improve the overall running time, they suggested using a weighted sampling $k \overset{\mathbf{w}}{\sim} [m]$ instead.

However, Eq. (10) holds only with an abuse of set operations. Precisely, repeated items are counted in their proof, e.g., $\{x, y, x\} = \{x, x, y\} \neq \{x, y\}$ and $\{z, x, z\} \cup z = \{z, x, z, z\} \neq \{z, x, z\}$. In this work, we adhere to set operations. Regarding Eq. (2), samples from $\mathcal{D}^{s-1}$ are tuples that may contain repeated items, but they are reduced to be sets.

Adhering to set operations, Eq. (10) is false. Take $\mathcal{D} \leftarrow \frac{1}{3}\delta_x + \frac{1}{3}\delta_y + \frac{1}{3}\delta_z$, $m \leftarrow 3$ as an example and assume $U(\emptyset) = 0$, one can verify that

$$\nu(z; \mathcal{D}, m, U) = \frac{1}{3}U(\{z\}) + \frac{5}{54}(U(\{x, z\}) + U(\{y, z\}) - U(\{x\}) - U(\{y\}))$$

$$+ \frac{2}{27}(U(\{x, y, z\}) - U(\{x, y\})),$$

$$\mathop{\mathbb{E}}_{k \overset{\mathcal{U}}{\sim} [m]} \mathop{\mathbb{E}}_{S \sim \mathcal{D}^{k-1}} [U(S \cup z) - U(S)] = \frac{1}{3}U(\{z\}) + \frac{4}{27}(U(\{x, z\}) + U(\{y, z\}) - U(\{x\}) - U(\{y\}))$$

$$+ \frac{2}{27}(U(\{x, y, z\}) - U(\{x, y\})).$$

Therefore, we refer to Eq. (2) in accordance with set operations as distributional values.

## B    PROOFS FOR THE PROPOSED ALGORITHMS

**Lemma 1.** *Let $\mathbf{J}_n \in \mathbb{R}^{n \times n}$ be the all-one matrix. The matrix $\mathbf{A} = a\mathbf{J}_n + b\mathbf{I}_n$ is invertible if and only if $na + b \neq 0$ and $b \neq 0$. Particularly, $\mathbf{A}^{-1} = -\frac{a}{b(na+b)}\mathbf{J}_n + \frac{1}{b}\mathbf{I}_n$.*

*Proof.* Observe that the eigenvalues of $\mathbf{J}_n$ are 0 (with $n - 1$ independent eigenvectors) and $n$, and thus the eigenvalues of $a\mathbf{J}_n + b\mathbf{I}_n$ are $b$ and $na + b$. Therefore, $a\mathbf{J}_n + b\mathbf{I}_n$ is invertible if and only if $b \neq 0$ and $na + b \neq 0$.

A priori is that $\mathbf{A}^{-1}$ admits the form of $x\mathbf{J}_n + y\mathbf{I}_n$. Using $\mathbf{A}\mathbf{A}^{-1} = \mathbf{I}_n$, we have the equation

$$(na + b)x + (a + b)y = 1,$$
$$(na + b)x + ay = 0,$$

which leads to that $x = -\frac{a}{b(na+b)}$ and $y = \frac{1}{b}$.  □

**Proposition 1.** *Define, for every $n \geq 1$, $U^n \in \mathcal{G}^n$ and $i \in [n]$,*

$$\mathcal{R}_i(U^n) = \mathcal{R}_i(U^n; \mathcal{P}) = \sum_{\emptyset \subsetneq S \subsetneq [n]} m_s^n \cdot \mathbb{1}_{i \in S} \cdot U^n(S) \tag{4}$$

*where $m_s^n = m_s^n(\mathcal{P}) = p_s^n + p_{s+1}^n$ for every $s \in [n-1]$. Then, for every $n \geq 1$ and $U^n \in \mathcal{G}^n$, $\mathcal{R}(U^n)$ and $\phi(U^n)$ produce the same ranking. Precisely, $\mathcal{R}(U^n) = \phi(U^n) + g(U^n)\mathbf{1}_n$ where $g(U^n) = g(U^n; \mathcal{P}) \in \mathbb{R}$. As a side note, it holds for any $\mathbf{p}^n \in \mathbb{R}^n$.*

*Proof.*

$$\begin{aligned} \phi_i(U^n) &= \sum_{S \subseteq [n] \setminus i} p_{s+1}^n (U^n(S \cup i) - U^n(S)) \\ &= \sum_{S \subsetneq [n]: \, i \in S} m_s^n U^n(S) + \left[ p_n^n U^n([n]) - \sum_{S \subsetneq [n]} p_{s+1}^n U^n(S) \right] \\ &= \sum_{\emptyset \subsetneq S \subsetneq [n]} m_s^n \cdot \mathbb{1}_{i \in S} \cdot U^n(S) + g(U^n). \end{aligned} \tag{11}$$

□

**Proposition 2.** *Consider the problem*

$$\arg\min_{\mathbf{v} \in \mathbb{R}^n} \sum_{\emptyset \subsetneq S \subsetneq [n]} m_s^n \cdot \left( U^n(S) - \sum_{i \in S} v_i \right)^2, \tag{5}$$

*its uniquely optimal solution shares the same ranking as $\phi(U^n)$. As a side note, it holds true for any $\mathbf{p}^n \in \mathbb{R}^n$ that produces a non-negative weight vector $\mathbf{m}^n$ with $\sum_{s=1}^{n-1} m_s^n > 0$.*

*Proof.* Let $\mathbf{J}_n \in \mathbb{R}^{n \times n}$ be the all-one matrix, and $\mathbf{I}_n \in \mathbb{R}^{n \times n}$ be the identity matrix. Since the problem (5) is convex, its optimal solution can be obtained by letting its derivative equal 0, which yields

$$\mathbf{A}\mathbf{v}^* = \mathbf{b}$$

where $\mathbf{A} = e\mathbf{J}_n + d\mathbf{I}_n$ with $e = \sum_{s=2}^{n-1} \binom{n-2}{s-2} m_s^n$ and $d = \left( \sum_{s=1}^{n-1} \binom{n-1}{s-1} m_s^n \right) - e,$ \tag{12}

and $\mathbf{b}_i = \sum_{S \subsetneq [n]: \, i \in S} m_s^n \cdot U^n(S)$ for every $i \in [n]$.

Specifically,

$$
\begin{aligned}
d &= m_1^n + \sum_{s=2}^{n-1} \left( \binom{n-2}{s-1} + \binom{n-2}{s-2} \right) m_s^n - \sum_{s=2}^{n-1} \binom{n-2}{s-2} m_s^n = \sum_{s=1}^{n-1} \binom{n-2}{s-1} m_s^n \\
&= \sum_{s=1}^{n-1} \binom{n-2}{s-1} p_s^n + \sum_{s=1}^{n-1} \binom{n-2}{s-1} p_{s+1}^n = p_1^n + \sum_{s=2}^{n-1} \binom{n-2}{s-1} p_s^n + \sum_{s=2}^{n-1} \binom{n-2}{s-2} p_s^n + p_n^n \\
&= \sum_{s=1}^{n} \binom{n-1}{s-1} p_s^n = 1.
\end{aligned}
$$

By Lemma 1, $\mathbf{A}^{-1} = -\frac{e}{e \cdot n + 1} \mathbf{J}_n + \mathbf{I}_n$, which implies the uniqueness of the optimal solution. Therefore,

$$
\mathbf{v}^* = \mathbf{A}^{-1}\mathbf{b} = \mathbf{b} + c\mathbf{1}_n \tag{13}
$$

where $c = \frac{-e}{e \cdot n + 1} \sum_{i=1}^{n} b_i$. According to Eq. (4), we have $\mathbf{b} = \mathcal{R}(U^n)$. Therefore, $\mathbf{v}^*$ and $\phi(U^n)$ share the same ranking. Suppose we have $\mathbf{p}^n \in \mathbb{R}^n$ with $\sum_{s=1}^{n-1} m_s^n > 0$, then there is $\mathbf{v}^* = \frac{1}{d}\mathbf{b} + \hat{c}\mathbf{1}_n$ instead where $\hat{c} = \frac{-e}{e \cdot n + d} \sum_{i=1}^{n} b_i$. Since $d = \sum_{s=1}^{n-1} \binom{n-2}{s-1} m_s^n > 0$, the factor $\frac{1}{d}$ does not change the ranking of $\mathbf{b}$. $\qquad\square$

**Proposition 3.** *The uniquely optimal solution $\mathbf{v}^*$ to the problem*

$$
\operatorname*{arg\,min}_{\mathbf{v} \in \mathbb{R}^{n+1}} \sum_{\emptyset \subsetneq S \subsetneq [n+1]} p_s^n \cdot \left( U^n(S \cap [n]) - \sum_{i \in S} v_i \right)^2 \tag{6}
$$

*satisfies that $\phi_i(U^n) = v_i^* - v_{n+1}^*$ for every $i \in [n]$.*

*Proof.* For each $U^n \in \mathcal{G}^n$, define $\overline{U}^{n+1} \in \mathcal{G}^{n+1}$ by letting $\overline{U}^{n+1}(S) = U^n(S \cap [n])$. Substituting $\overline{U}^{n+1}(S)$ for $U^n(S \cap [n])$ in the problem (6), and reusing the Eq. (12) accordingly, we have

$$
\mathbf{v}^* = \mathbf{A}^{-1}\mathbf{b}
$$

$$
\text{where } b_i = p_n^n \cdot U^n([n]) + \sum_{S \subsetneq [n]:\, i \in S} (p_s^n + p_{s+1}^n) \cdot U^n(S) \quad \forall i \in [n],
$$

$$
\text{and } b_{n+1} = \sum_{S \subsetneq [n]} p_{s+1}^n \cdot U^n(S).
$$

Using Eq. (11), we obtain $\phi_i(U^n) = b_i - b_{n+1}$ for every $i \in [n]$. Meanwhile, by Eq. (13), we have $\mathbf{v}^* = \mathbf{b} + c\mathbf{1}_{n+1}$ for some $c \in \mathbb{R}$, and thus $v_i - v_{n+1} = b_i - b_{n+1} = \phi_i(U^n)$ for every $i \in [n]$. $\quad\square$

**Theorem 1.** *Consider the case when $B = [n]$ (recall that $D_{tr} \equiv [n]$) for the empirical distributional values $\varphi$ defined in Eq. (7). Let $\mathbf{d} \in \mathbb{R}^m$ be a probability vector that satisfies $d_s \propto \frac{n-s+1}{s} \omega_s$ for every $s \in [m]$, and $\mathbf{v}^*$ be the uniquely optimal solution to*

$$
\operatorname*{arg\,min}_{\mathbf{v} \in \mathbb{R}^{n+1}} \; \mathbb{E}_{s \sim \mathbf{d}[m]} \mathbb{E}_{S \sim \mathcal{U}_{\overline{\mathcal{B}}_s}} \left( U^n(S \cap [n]) - \sum_{i \in S} v_i \right)^2, \tag{8}
$$

*where $\overline{\mathcal{B}}_s = \{ R \subseteq [n+1] \mid |R| = s \}$. There is, for every $i \in [n]$,*

$$
C \cdot (v_i^* - v_{n+1}^*) = \varphi(i; \mathcal{B}, \boldsymbol{\omega}, U^n) \text{ where } C = \sum_{s=1}^{m} \frac{n-s+1}{n} \omega_s.
$$

*Additionally, setting $p_s^n \propto \binom{n}{s-1}^{-1} \omega_s$ if $s \in [m]$ and $0$ otherwise such that $\mathbf{p}^n$ defines a probabilistic value, $\hat{\mathbf{r}}$ obtained in Algorithm 2 meets that $\mathbb{E}[\hat{r}_i] - \mathbb{E}[\hat{r}_{n+1}] = \varphi(i; \mathcal{B}, \boldsymbol{\omega}, U^n)$ for every $i \in [n]$.*

*Proof.* Since $U^n(S \cup i) - U^n(S) = 0$ if $i \in S$, for each $i \in [n]$,

$$
\begin{aligned}
\varphi(i; \mathcal{B}, \boldsymbol{\omega}, U^n) &= \sum_{S \subseteq [n] \setminus i:\, s < m} \omega_{s+1} \binom{n}{s}^{-1} (U^n(S \cup i) - U^n(S)) \\
&= C \cdot \sum_{S \subseteq [n] \setminus i} q_{s+1} \cdot (U^n(S \cup i) - U^n(S))
\end{aligned}
$$

where $q_s = \binom{n}{s-1}^{-1} \frac{\omega_s}{C}$ if $s \in [m]$ and 0 otherwise, and $C = \sum_{s=1}^m \binom{n-1}{s-1} \binom{n}{s-1}^{-1} \omega_s = \sum_{s=1}^m \frac{n-s+1}{n} \omega_s$. On the other hand, the objective in the problem (8) can be rewritten as

$$\sum_{\emptyset \subsetneq S \subsetneq [n+1]:\ s \leq m} d_s \binom{n}{s}^{-1} \left( \overline{U}^{n+1}(S) - \sum_{i \in S} v_i \right)^2$$

where $\overline{U}^{n+1} \in \mathcal{G}^{n+1}$ is defined by letting $\overline{U}^{n+1}(S) = U^n(S \cap [n])$ for every $S \subseteq [n+1]$. Note that $d_s = \alpha \frac{n-s+1}{s} \omega_s$ for every $s \in [m]$ where $\alpha$ is a scalar that makes $\mathbf{d}$ a probability vector. By Proposition 3, for every $i \in [n]$,

$$v_i^* - v_{n+1}^* = \sum_{S \subseteq [n] \setminus i} \hat{q}_{s+1} \cdot (U^n(S \cup i) - U^n(S))$$

where $\hat{q}_s = \beta \cdot d_s \binom{n}{s}^{-1}$ for every $s \in [m]$ and 0 otherwise. Suffice it to show that $\hat{q}_s = q_s$ for every $s \in [m]$. Note that

$$1 = \sum_{s=1}^m \binom{n-1}{s-1} \hat{q}_s = \sum_{s=1}^m \binom{n-1}{s-1} \beta \cdot \alpha \frac{n-s+1}{s} \omega_s \binom{n}{s}^{-1} = \alpha \cdot \beta \cdot C,$$

and thus

$$q_s = \beta \cdot \alpha \binom{n}{s-1}^{-1} \omega_s = \beta \binom{n}{s-1}^{-1} \frac{s}{n-s+1} d_s = \beta \binom{n}{s}^{-1} d_s = \hat{q}_s.$$

For the side note, note that

$$L \cdot (\mathbb{E}[\hat{r}_i] - \mathbb{E}[\hat{r}_{n+1}]) = v_i^* - v_{n+1}^*$$

where

$$L = \sum_{s=1}^n \frac{s}{n+1} \binom{n+1}{s} q_s = \sum_{s=1}^m \frac{s}{n+1} \binom{n+1}{s} \binom{n}{s-1}^{-1} \frac{\omega_s}{C} = \frac{1}{C}.$$

$\square$

## C   PROOFS FOR CONVERGENCES

**Proposition 5.** *Assume that $\|U^n\|_\infty \leq u$ for every $U^n \in \bigcup_{n \geq 1} \mathcal{G}^n$, any estimator based on the sampling lift strategy requires $O(\frac{n^2}{\epsilon^2} \log \frac{n}{\delta})$ utility evaluations to achieve $P(\|\hat{\phi}(U^n) - \phi(U^n)\|_2 \geq \epsilon) \leq \delta$.*

The sampling lift strategy refers to any estimator designed according to

$$\phi_i(U^n) = \sum_{S \subseteq [n] \setminus i} p_{s+1}^n (U^n(S \cup i) - U^n(S)) = \mathbb{E}_S[U^n(S \cup i) - U^n(S)]$$

for every $i \in [n]$ as $\sum_{s=1}^n \binom{n-1}{s-1} p_s^n = 1$. Precisely, $U^n(S \cup i) - U^n(S)$ is a random variable following a certain probability distribution, denoted by $P_i$, on all subsets $S \subseteq [n] \setminus i$. Let $T$ be the total number of samples and $\mathcal{S} = \{(S_1^1, S_1^2, \ldots, S_1^n), (S_2^1, S_2^2, \ldots, S_2^n), \ldots, (S_T^1, S_T^2, \ldots, S_T^n)\}$ be a list of $T$ samples where each $\mathcal{S}^i = \{S_1^i, S_2^i, \ldots, S_T^i\}$ are sampled from $P_i$. Then, the estimator based on the sampling lift strategy is $\hat{\phi}_i(U^n) = \frac{1}{T} \sum_{k=1}^T (U^n(S_k^i \cup i) - U^n(S_k^i))$. There are two slightly different implementations: i) sampling each $\mathcal{S}^i$ independently, and ii) sampling $\mathcal{S}^i$ for some $i \in [n]$, and then reusing all samples in $\mathcal{S}^i$ for every other $j \in [n]$ by swapping $i$ and $j$, the latter of which is our implementation. Nevertheless, the proof below works for both of them. Besides, the proof is adapted from (Wang and Jia 2023, Theorem 4.8) where it is stated for the Banzhaf value. Nevertheless, we point out that it applies to all probabilistic values.

*Proof.* For simplicity, we write $\phi = \phi(U^n)$ and $\hat{\phi} = \hat{\phi}(U^n)$. Fix an $i \in [n]$, since $\mathbb{E}[\hat{\phi}_i] = \phi_i$, by the Hoeffding's inequality,

$$P(|\hat{\phi}_i - \phi_i| \geq \epsilon) \leq 2 \exp(-\frac{T\epsilon^2}{2u^2}).$$

Then,

$$P(\|\hat{\phi} - \phi\|_2 \geq \epsilon) \leq P(\bigcup_{i \in [n]} |\hat{\phi}_i - \phi_i| \geq \frac{\epsilon}{\sqrt{n}}) \leq 2n \exp(-\frac{T\epsilon^2}{2nu^2})$$

Letting $\delta \geq 2n \exp(-\frac{T\epsilon^2}{2nu^2})$ leads to $T \geq \frac{2nu^2}{\epsilon^2} \log \frac{2n}{\delta}$. Since each sample $(S_k^1, S_k^2, \ldots, S_k^n)$ invokes $2n$ utility evaluations, it requires $O(\frac{n^2}{\epsilon^2} \log \frac{n}{\delta})$ utility evaluations to achieve an $(\epsilon, \delta)$-approximation. $\square$

**Proposition 4.** *Assume that $\|U^n\|_\infty \leq u$ for every $U^n \in \bigcup_{n \geq 1} \mathcal{G}^n$. We have the following results: i) GELS requires $O(\frac{\tau(n)n}{\epsilon^2} \log \frac{n}{\delta})$ utility evaluations to achieve an $(\epsilon, \delta)$-approximation, i.e., $P(\|\hat{\phi}(U^n) - \phi(U^n)\|_2 \geq \epsilon) \leq \delta$; ii) for GELS-R estimator, it requires $O(\frac{\kappa(n)n}{\epsilon^2} \log \frac{n}{\delta})$ utility evaluations instead; iii) plus, the corresponding convergence of GELS-Shapley is $O(\frac{n}{\epsilon^2} \log(n)^2 \log \frac{n}{\delta})$.*

*Proof.* The proof is adapted from (Wang and Jia 2023, Theorem 4.9). First, we prove the convergence for GELS-R. Note that it is an unbiased estimator of

$$\mathbf{r} = \left(\sum_{s=1}^{n-1} \binom{n-1}{s-1} (p_s^n + p_{s+1}^n)\right)^{-1} \mathcal{R}(U^n). \tag{14}$$

Let $\gamma(n) = \frac{\sum_{s=1}^{n-1} \binom{n-1}{s-1}(p_s^n + p_{s+1}^n)}{\sum_{s=1}^{n-1} \binom{n}{s}(p_s^n + p_{s+1}^n)}$. For convenience, we write $\mathcal{S} = \{S_1, S_2, \ldots, S_T\}$ that contains all sampled subsets, and $T_i = |\{S \in \mathcal{S} \mid i \in S\}|$ for every $i \in [n]$. For every $i \in [n]$, define

$$\bar{r}_i = \frac{1}{\gamma(n)T} \sum_{S \in \mathcal{S}: i \in S} U^n(S).$$

Then, we have

$$|\hat{r}_i - \bar{r}_i| = \left|\left(\frac{1}{T_i} - \frac{1}{\gamma(n)T}\right) \sum_{S \in \mathcal{S}: i \in S} U^n(S)\right| \leq \frac{u}{\gamma(n)T} |\gamma(n)T - T_i|.$$

Note that this inequality $|\hat{r}_i - \bar{r}_i| \leq \frac{u}{\gamma(n)T} |\gamma(n)T - T_i|$ still holds when $T_i = 0$. Since $T_i \sim$ binomial$(T, \gamma(n))$, by the Hoeffding's inequality, there is

$$P(|T_i - \gamma(n)T| \geq \Delta) \leq 2 \exp(-\frac{2\Delta^2}{T}).$$

Therefore, $|\hat{r}_i - \bar{r}_i| < \frac{u\Delta}{\gamma(n)T}$ provided that $|T_i - \gamma(n)T| < \Delta$. Since $\bar{r}_i = \frac{1}{T} \sum_{S \in \mathcal{S}} \beta(S, n) U^n(S)$ where $\beta(S, n) = \gamma(n)^{-1}$ if $i \in S$ and 0 otherwise, and $\mathbb{E}[\bar{r}_i] = r_i$, using the Heoffding's inequality again yields

$$P(|\bar{r}_i - r_i| \geq \sigma) \leq 2 \exp(-\frac{\gamma(n)^2 T \sigma^2}{2u^2})$$

Therefore,

$$P(|\hat{r}_i - r_i| \geq \epsilon) = P(|\hat{r}_i - r_i| \geq \epsilon \cap |T_i - \gamma(n)T| < \Delta) + P(|\hat{r}_i - r_i| \geq \epsilon \cap |T_i - \gamma(n)T| \geq \Delta)$$

$$\leq P(|\hat{r}_i - r_i| \geq \epsilon \mid |T_i - \gamma(n)T| < \Delta) + 2 \exp(-\frac{2\Delta^2}{T})$$

$$\leq P(|\bar{r}_i - r_i| \geq \epsilon - \frac{u\Delta}{\gamma(n)T} \mid |T_i - \gamma(n)T| < \Delta) + 2 \exp(-\frac{2\Delta^2}{T})$$

$$\leq \frac{P(|\bar{r}_i - r_i| \geq \epsilon - \frac{u\Delta}{\gamma(n)T})}{1 - 2 \exp(-\frac{2\Delta^2}{T})} + 2 \exp(-\frac{2\Delta^2}{T})$$

$$\leq \frac{2 \exp(-\frac{\gamma(n)^2 T\left(\epsilon - \frac{u\Delta}{\gamma(n)T}\right)^2}{2u^2})}{1 - 2 \exp(-\frac{2\Delta^2}{T})} + 2 \exp(-\frac{2\Delta^2}{T})$$

$$\leq 3 \exp(-\frac{\gamma(n)^2 T \left(\epsilon - \frac{u\Delta}{\gamma(n)T}\right)^2}{2u^2}) + 2 \exp(-\frac{2\Delta^2}{T})$$

where $1 - 2\exp(-\frac{2\Delta^2}{T}) \geq \frac{2}{3}$ provided that $T$ is sufficiently large. The next step is to determine $\Delta$ by solving the equation $-\frac{\gamma(n)^2 T\left(\epsilon - \frac{u\Delta}{\gamma(n)T}\right)^2}{2u^2} = -\frac{2\Delta^2}{T}$, which yields $\Delta = \frac{\gamma(n)T\epsilon}{3u}$. Note that this solution gives $\epsilon - \frac{u\Delta}{\gamma(n)T} = \frac{2\epsilon}{3} > 0$ and $\frac{2\Delta^2}{T} = \frac{2\gamma(n)^2 T\epsilon^2}{9u^2}$. Besides, the inequality $1 - 2\exp(-\frac{2\Delta^2}{T}) \geq \frac{2}{3}$ leads to $T \geq \frac{9\log(6)u^2}{2\gamma(n)^2\epsilon^2}$. Eventually, we have

$$P(|\hat{r}_i - r_i| \geq \epsilon) \leq 5\exp(-\frac{2\gamma(n)^2 T\epsilon^2}{9u^2}) \tag{15}$$

provided that $T \geq \frac{9\log(6)u^2}{2\gamma(n)^2\epsilon^2}$. Then,

$$P(\|\hat{\mathbf{r}} - \mathbf{r}\|_2 \geq \epsilon) \leq P(\bigcup_{i \in [n]} |\hat{r}_i - r_i| \geq \frac{\epsilon}{\sqrt{n}}) \leq 5n\exp(-\frac{2\gamma(n)^2 T\epsilon^2}{9u^2 n}). \tag{16}$$

Solving $5n\exp(-\frac{2\gamma(n)^2 T\epsilon^2}{9u^2 n}) \leq \delta$ leads to $T \geq \frac{9u^2 n}{2\gamma(n)^2\epsilon^2}\log\frac{5n}{\delta}$, and thus GELS-R requires

$$\max(\frac{9\log(6)u^2}{2\gamma(n)^2\epsilon^2}, \frac{9u^2 n}{2\gamma(n)^2\epsilon^2}\log\frac{5n}{\delta}) = O(\frac{\kappa(n)n}{\epsilon^2}\log\frac{n}{\delta})$$

utility evaluations to achieve an $(\epsilon, \delta)$-approximation where $\kappa(n) = \gamma(n)^{-2}$.

Next, we prove the convergence for GELS. For simplicity, we write $\boldsymbol{\phi} = \phi(U^n)$ and $\mathbf{h} = \left(\sum_{s=1}^n \binom{n}{s-1}p_s^n\right)^{-1}\mathcal{R}(\overline{U}^{n+1})$; see Eq. (14). By Eq. (15), there is, for every $i \in [n+1]$,

$$P(|\hat{h}_i - h_i| \geq \epsilon) \leq 5\exp(-\frac{2\widetilde{\gamma}(n)^2 T\epsilon^2}{9u^2})$$

provided that $T \geq \frac{9\log(6)u^2}{2\widetilde{\gamma}(n)^2\epsilon^2}$ where $\widetilde{\gamma}(n) = \frac{\sum_{s=1}^n \binom{n}{s-1}p_s^n}{\sum_{s=1}^n \binom{n+1}{s}p_s^n}$. Therefore, for every $i \in [n]$,

$$P(|(\hat{h}_i - \hat{h}_{n+1}) - (h_i - h_{n+1})| \geq \epsilon) \leq P(|\hat{h}_i - h_i| \geq \frac{\epsilon}{2} \cup |\hat{h}_{n+1} - h_{n+1}| \geq \frac{\epsilon}{2})$$

$$\leq 10\exp(-\frac{\widetilde{\gamma}(n)^2 T\epsilon^2}{18u^2}).$$

let $\chi(n) = \sum_{s=1}^n \binom{n}{s-1}p_s^n$ and $\eta(n) = \sum_{s=1}^n \binom{n+1}{s}p_s^n$, and note that $\widetilde{\gamma}(n) = \frac{\chi(n)}{\eta(n)}$. As argued in Remark 1, $\mathcal{R}_i(\overline{U}^{n+1}) - \mathcal{R}_{n+1}(\overline{U}^{n+1}) = \phi_i(U^n)$ for every $i \in [n]$, and thus $\chi(n)(h_i - h_{n+1}) = \phi_i(U^n)$ for every $i \in [n]$. So, for every $i \in [n]$,

$$P(|\hat{\phi}_i - \phi_i| \geq \epsilon) = P(|(\hat{h}_i - \hat{h}_{n+1}) - (h_i - h_{n+1})| \geq \frac{\epsilon}{\chi(n)}) \leq 10\exp(-\frac{T\epsilon^2}{18u^2\eta(n)^2}).$$

Therefore,

$$P(\|\hat{\boldsymbol{\phi}} - \boldsymbol{\phi}\|_2 \geq \epsilon) \leq P(\bigcup_{i \in [n]} |\hat{\phi}_i - \phi_i| \geq \frac{\epsilon}{\sqrt{n}}) \leq 10n\exp(-\frac{T\epsilon^2}{18u^2 n\eta(n)^2}).$$

Solving $\delta \geq 10n\exp(-\frac{T\epsilon^2}{18u^2 n\eta(n)^2})$ yields $T \geq \frac{18u^2 n\eta(n)^2}{\epsilon^2}\log\frac{10n}{\delta}$. To conclude, GELS requires

$$\max(\frac{9\log(6)u^2}{2\widetilde{\gamma}(n)^2\epsilon^2}, \frac{18u^2 n\eta(n)^2}{\epsilon^2}\log\frac{10n}{\delta}) = O(\frac{\tau(n)n}{\epsilon^2}\log\frac{n}{\delta})$$

utility evaluations to achieve an $(\epsilon, \delta)$-approximation where $\tau(n) = \eta(n)^2$. Note that $\widetilde{\gamma}(n)^{-1} \leq \eta(n)$ as $\chi(n) = \sum_{s=1}^n \binom{n}{s-1}\binom{n-1}{s-1}^{-1}\binom{n-1}{s-1}p_s^n = \sum_{s=1}^n \frac{n}{n-s+1}\binom{n-1}{s-1}p_s^n \geq 1$.

Last, we prove the convergence for GELS-Shapley. Reusing Eq. (14), since $p_s^n = \frac{(s-1)!(n-s)!}{n!}$ for the Shapley value, it is $H_{n-1} \cdot \mathbf{r} = \mathcal{R}(U^n)$ where $H_{n-1} = \sum_{s=1}^{n-1}\binom{n-1}{s-1}(p_s^n + p_{s+1}^n) = \sum_{s=1}^{n-1}\frac{1}{s}$. By Eq. (16), there is

$$P(H_{n-1} \cdot \|\hat{\mathbf{r}} - \mathbf{r}\|_2 \geq \epsilon) \leq 5n\exp\left(-\frac{2T\epsilon^2}{9u^2 n\eta(n-1)^2}\right)$$

provided that $T \geq \frac{9\log(6)u^2}{2\gamma(n)^2\epsilon^2}$. Note that $\sum_{s=1}^{n-1}\binom{n}{s}(p_s^n + p_{s+1}^n) = \sum_{s=1}^{n-1}\binom{n}{s}p_s^{n-1} = \eta(n-1)$.

For convenience, let $\phi^{Sh}$ be the corresponding Shapley value. As discussed in Remark 2,

$$\phi^{Sh} = H_{n-1} \cdot \mathbf{r} + \frac{U^n([n]) - U^n(\emptyset) - \sum_{s=1}^n H_{n-1} \cdot r_i}{n}.$$

The procedure of casting $H_{n-1} \cdot \mathbf{r}$ into $\phi^{Sh}$ is called additive-efficient-normalization (Ruiz et al. 1998, Definition 11). Particularly, we write

$$\hat{\phi}^{Sh} = H_{n-1} \cdot \hat{\mathbf{r}} + \frac{U^n([n]) - U^n(\emptyset) - \sum_{s=1}^n H_{n-1} \cdot \hat{r}_i}{n}.$$

As pointed out by Jethani et al. (2022, Appendix B), the additive-efficient-normalization is just an orthogonal projection, and therefore we have

$$\|\hat{\phi}^{Sh} - \phi^{Sh}\|_2 \leq \|H_{n-1} \cdot \hat{\mathbf{r}} - H_{n-1} \cdot \mathbf{r}\|_2,$$

which suggest

$$P(\|\hat{\phi}^{Sh} - \phi^{Sh}\|_2 \geq \epsilon) \leq P(\|H_{n-1} \cdot \hat{\mathbf{r}} - H_{n-1} \cdot \mathbf{r}\|_2 \geq \epsilon) \leq 5n\exp\left(-\frac{2T\epsilon^2}{9u^2 n\eta(n-1)^2}\right).$$

Solving $\delta \geq 5n\exp\left(-\frac{2T\epsilon^2}{9u^2 n\eta(n-1)^2}\right)$ yields $T \geq \frac{9u^2 n\eta(n-1)^2}{2\epsilon^2}\log\frac{5n}{\delta}$. Since $\gamma(n)^{-1} \leq \eta(n-1)$, we eventually have that GELS-Shapley requires $O(\frac{n\tau(n)}{\epsilon^2}\log\frac{n}{\delta})$ utility evaluations to achieve an $(\epsilon, \delta)$-approximation where $\tau(n) = \eta(n)^2 > \eta(n-1)^2$. Note that i) for the Shapley value, $\eta(n) = \sum_{s=1}^n \binom{n+1}{s}\binom{n-1}{s-1}^{-1}\frac{1}{n} = \sum_{s=1}^n \frac{n+1}{s(n+1-s)} = \sum_{s=1}^n \left(\frac{1}{s} + \frac{1}{n+1-s}\right) = 2H_n \in \log n$, and thus $\eta(n-1) < \eta(n)$; and ii) $\gamma(n)^{-1} \leq \eta(n-1)$ as $\gamma(n) \cdot \eta(n-1) = H_{n-1} \geq 1$. $\qquad\square$

## D  LEAST SQUARE VALUES

In this appendix, we demonstrate how we obtained proposition 2 at the very beginning without knowing proposition 1.

**Definition 1** (Least Square Values (Ruiz et al. 1998, Definition 5)). *Suppose a non-negative non-zero vector $\mathbf{m} \in \mathbb{R}^{n-1}$ is given, a least square value $\xi(U^n; \mathbf{m}) \in \mathbb{R}^n$ is defined to be the uniquely optimal solution to the problem*

$$\arg\min_{\mathbf{v} \in \mathbb{R}^n} \sum_{\emptyset \subsetneq S \subsetneq [n]} m_s \left(U^n(S) - U^n(\emptyset) - \sum_{i \in S} v_i\right)^2 \tag{17}$$
$$s.t. \sum_{i=1}^n v_i = U^n([n]) - U^n(\emptyset).$$

Specifically, Ruiz et al. (1998, Theorem 8) developed a system of axioms that uniquely characterizes the family of least square values. Moreover, its relationship with the family of probabilistic values was also revealed.

**Proposition 6** (Ruiz et al. 1998, Theorem 12). *For each vector of weights $\mathbf{p}^n \in \mathbb{R}^n$ that satisfies $\sum_{s=1}^n \binom{n-1}{s-1}p_s^n = 1$, define $\mathbf{m} \in \mathbb{R}^{n-1}$ by letting $m_s = p_s^n + p_{s+1}^n$, there exists some function $f : \mathcal{G}^n \to \mathbb{R}$ such that*

$$\xi_i(U^n; \mathbf{m}) = \sum_{S \subseteq [n]\setminus i} p_s^n \cdot (U^n(S \cup i) - U^n(\emptyset)) + f(U^n)$$

*for every $U^n \in \mathcal{G}^n$ and $i \in [n]$.*

As proposed by Ruiz et al. (1998, Definition 11), for each probabilistic value $\phi(U^n)$, its additive-efficient-normalization $\overline{\phi}$ is defined to be

$$\overline{\phi}(U^n) = \phi(U^n) + \frac{1}{n}(U^n([n]) - U^n(\emptyset) - \sum_{i \in [n]} \phi_i(U^n)).$$

Using Proposition 6, one can verify that $\xi(U^n; \mathbf{m}) = \overline{\phi}(U^n)$ and $f(U^n) = \frac{1}{n}(U^n([n]) - U^n(\emptyset) - \sum_{i \in [n]} \phi_i(U^n))$. Particularly, we notice that the constraint of the problem (17) and the term $-U^n(\emptyset)$ in the objective can be removed if it is the ranking that matters.

**Proposition 7.** *Suppose a utility function $U^n \in \mathcal{G}^n$ and a non-negative non-zero vector $\mathbf{m} \in \mathbb{R}^{n-1}$ are given, the uniquely optimal solution $\mathbf{u}^*$ to the problem*

$$\underset{\mathbf{u} \in \mathbb{R}^n}{\arg\min} \sum_{\emptyset \subsetneq S \subsetneq [n]} m_s \left( U^n(S) - \sum_{i \in S} u_i \right)^2 \tag{18}$$

*satisfies that there exists some $c \in \mathbb{R}$ such that*

$$\mathbf{u}^* = \xi(U^n; \mathbf{m}) + c\mathbf{1}_n.$$

*Proof.* Since the optimization problem (17) is convex, it can be solved using the KKT condition. Specifically, by introducing a dual variable $\lambda \in \mathbb{R}$, there is

$$\mathbf{v}^\top \mathbf{A}\mathbf{v} - 2\mathbf{b}^\top \mathbf{v} + \sum_{\emptyset \subsetneq S \subsetneq [n]} m_s \left( U^n(S) - U^n(\emptyset) \right)^2 - 2\lambda(U^n([n]) - U^n(\emptyset) - \sum_{i=1}^{n} \phi_i)$$

where $A_{ij} = \begin{cases} \sum_{s=1}^{n-1} \binom{n-1}{s-1} m_s, & i = j \\ \sum_{s=2}^{n-1} \binom{n-2}{s-2} m_s, & i \neq j \end{cases}$, and $b_i = \sum_{S \subsetneq [n]:\, i \in S} m_s \left( U^n(S) - U^n(\emptyset) \right) \quad \forall i \in [n]$.

Letting its derivative equal 0 yields

$$\mathbf{A}\mathbf{v} - \mathbf{b} + \lambda\mathbf{1}_n = 0 \text{ and } \mathbf{1}_n^\top \mathbf{v} = U^n([n]) - U^n(\emptyset),$$

which leads to the uniquely optimal solution

$$\mathbf{v}^* = \mathbf{A}^{-1} \left( \mathbf{b} - \frac{\mathbf{1}_n^\top \mathbf{A}^{-1}\mathbf{b} - U^n([n]) + U^n(\emptyset)}{\mathbf{1}_n^\top \mathbf{A}^{-1}\mathbf{1}_n} \mathbf{1}_n \right). \tag{19}$$

Specifically,

$$\sum_{s=1}^{n-1} \binom{n-1}{s-1} m_s - \sum_{s=2}^{n-1} \binom{n-2}{s-2} m_s$$

$$= m_1 + \sum_{s=2}^{n-1} \left( \binom{n-2}{s-1} + \binom{n-2}{s-2} \right) m_s - \sum_{s=2}^{n-1} \binom{n-2}{s-2} m_s = \sum_{s=1}^{n-1} \binom{n-2}{s-1} m_s > 0,$$

by lemma 1, the existence of $\mathbf{A}^{-1}$ is guaranteed. On the other hand, letting the derivative of the objective in the problem (18) equal 0 gives

$$\mathbf{u}^* = \mathbf{A}^{-1}(\mathbf{b} + t\mathbf{1}_n) \text{ where } t = U^n(\emptyset) \cdot A_{11}.$$

Therefore, we eventually have, for some $c \in \mathbb{R}$,

$$\mathbf{u}^* = \mathbf{v}^* + c\mathbf{1}_n = \xi(U^n; \mathbf{m}) + c\mathbf{1}.$$

for some $c \in \mathbb{R}$. $\qquad \square$

**Remark 5.** *To conclude, Proposition 2 is derived from Propositions 6 and 7.*

# E    GENERALIZED ARM

As proposed by Kolpaczki et al. (2023), the Shapley value can be rewritten as, for every $U^n \in \mathcal{G}^n$ and $i \in [n]$,

$$\phi_i^{Sh}(U^n) = \mathbb{E}_{S \sim P_{ARM}^+ | i \in S}[U^n(S)] - \mathbb{E}_{S \sim P_{ARM}^- | i \notin S}[U^n(S)]$$

where $P_{ARM}^+(S) = \frac{1}{s \cdot H} \binom{n}{s}^{-1}$ for every $S \in \mathcal{S}^+ = \{\emptyset \subsetneq S \subseteq [n]\}$ and $P_{ARM}^-(S) = \frac{1}{(n-s) \cdot H} \binom{n}{s}^{-1}$ for every $S \in \mathcal{S}^- = \{S \subsetneq [n]\}$; $H = \sum_{s=1}^{n} \frac{1}{s}$.

The estimator based on this formula is called the approximation-without-requiring-marginal estimator. We found that this methodology can be easily adapted for every other probabilistic value, which is summarized in the below.

**Proposition 8.** *For every $U^n \in \mathcal{G}^n$ and $i \in [n]$,*

$$\phi_i(U^n) = \mathbb{E}_{S \sim P^+_{ARM}|i \in S}[U^n(S)] - \mathbb{E}_{S \sim P^-_{ARM}|i \notin S}[U^n(S)]$$

*where $P^+_{ARM}(S) \propto p_s$ for every $S \in \mathcal{S}^+$ and $P^-_{ARM}(S) \propto p_{s+1}$ for every $S \in \mathcal{S}^-$.*

*Proof.* Precisely, $p^+_s = \alpha \cdot p_s$ and $p^-_s = \beta \cdot p_{s+1}$ such that $\sum_{s=1}^{n} \binom{n}{s} p^+_s = 1$ and $\sum_{s=0}^{n-1} \binom{n}{s} p^-_{s+1} = 1$. Let $S^+$ and $S^-$ be the corresponding random samples from $\mathcal{S}^+$ and $\mathcal{S}^-$, respectively. Fix an $i \in [n]$, observe that,

$$P(i \in S^+) = \sum_{s=1}^{n} P(i \in S^+ \mid |S^+| = s) \cdot P(|S^+| = s) = \sum_{s=1}^{n} \binom{n-1}{s-1} \binom{n}{s}^{-1} \cdot \binom{n}{s} p^+_s$$

$$= \alpha \sum_{s=1}^{n} \binom{n-1}{s-1} p_s = \alpha.$$

Therefore,

$$P(S^+ \mid i \in S^+) = \frac{P(S^+, i \in S^+)}{P(i \in S^+)} = \frac{\alpha \cdot p_s}{\alpha} = p_s.$$

Similarly,

$$P(i \notin S^-) = \sum_{s=0}^{N-1} P(i \notin S^- \mid |S^-| = s) \cdot P(|S^-| = s) = \sum_{s=0}^{n-1} \binom{n-1}{s} \binom{n}{s}^{-1} \cdot \binom{n}{s} p^-_s$$

$$= \beta \sum_{s=0}^{n-1} \binom{n-1}{s} p_{s+1} = \beta,$$

which leads to

$$P(S^- \mid i \notin S^-) = \frac{P(S^-, i \notin S^-)}{P(i \notin S^-)} = \frac{\beta \cdot p_{s+1}}{\beta} = p_{s+1}.$$

$\square$

## F  PRACTICAL PROBABILISTIC VALUES

As provided by Dubey et al. (1981), each semi-value, which is a subfamily of probabilistic values, can be expressed as, for every $U^n \in \mathcal{G}^n$ and $i \in [n]$,

$$\phi_i(U^n) = \phi_i(U^n; \mu) = \sum_{S \subseteq [n] \setminus i} \left( \int_0^1 t^s (1-t)^{n-1-s} d\mu(t) \right) (U^n(S \cup i) - U^n(S))$$

where $\mu$ is any probability measure on the closed interval $[0, 1]$. In other words, in view of Eq. (1), there is $p^n_s = \int_0^1 t^{s-1}(1-t)^{n-s} d\mu(t)$.

To the best of our knowledge, practical semi-values, i.e., those ever studied in the previous references, include the Banzhaf value (Wang and Jia 2023) and the Beta Shapley values with $\alpha, \beta \geq 1$ (Kwon and Zou 2022a; Kwon and Zou 2022b). Note that Beta$(1, 1)$ is exactly the Shapley value. For the Banzhaf value, the corresponding $\mu$ is the Dirac delta distribution $\delta_{0.5}$, which leads to $p^n_s = 2^{-(n-1)}$. For Beta$(\alpha, \beta)$, the corresponding probability density function for $\mu$ is $\propto t^{\beta-1}(1-t)^{\alpha-1}$, and thus $p^n_s = \frac{\Gamma(\alpha+\beta)}{\Gamma(\alpha)\Gamma(\beta)} \cdot \frac{\Gamma(\beta+s-1)\Gamma(\alpha+n-s)}{\Gamma(\alpha+\beta+n-1)}$. Specifically, it is $p^n_s = \frac{(s-1)!(n-s)!}{n!}$ for Beta$(1, 1)$, i.e., the Shapley value.

## G  PRACTICAL ASPECT OF FASTER ESTIMATORS FOR PROBABILISTIC VALUES

As far as we know, in feature attribution, FastSHAP is the only framework for training semi-value-based explainers (Jethani et al. 2022), which is based on the least square regression Eq. (3) specific

to the Shapley value. Recently, Kwon and Zou (2022b) showed that other candidates of the Beta Shapley values tend to perform better than the Shapley value in feature attribution. Therefore, one may ask how to cast other probabilistic values into optimization, which is answered by Propositions 1 and 3. Though AME proposed by Lin et al. (2022) provides an alternative way to achieve this goal, it is restricted to a subfamily of semi-values, which does not include, e.g., the Shapley value and Beta$(4, 1)$ used in our experiments. Besides, as shown in our experiments of comparing convergences, on Beta$(2, 2)$, the induced estimator of AME does not rival our GELS-R and GELS estimators, which are derived from solving the least square regressions (5) and (6).

Recall that the optimization problem used by FastSHAP is

$$\mathbb{E}_{z \in \mathcal{Z}} \left[ \sum_{\emptyset \subsetneq S \subsetneq [d]} \frac{d-1}{\binom{d}{s} s(d-s)} \left( U_z(S) - U_z(\emptyset) - \mathbf{1}_S^\top \phi(z; \boldsymbol{\theta}) \right)^2 \right]$$

where $d$ is the number of features (a counterpart of $n$), $\phi(z; \boldsymbol{\theta}) \in \mathbb{R}^d$ is a trainable explainer parameterized by $\boldsymbol{\theta}$, $U_z$ is a utility function based on the data point $z = (\mathbf{x}, y)$ where $\mathbf{x}$ and $y$ represent the features and label, respectively, and $\mathbf{1}_S \in \mathbb{R}^d$ is defined by $\mathbf{1}_S(i) = 1$ if $i \in S$ and 0 otherwise.

Assume $\{S\}_{\emptyset \subsetneq S \subsetneq [d]}$ is ordered so that each utility function $U_z$ can be treated as a vector $\mathbf{U}_z \in \mathbb{R}^{2^d-2}$. Besides, let $\mathbf{W} \in \mathbb{R}^{(2^d-2) \times (2^d-2)}$ be a diagonal matrix such that $W(S, S) = \frac{d-1}{\binom{d}{s} s(d-s)}$, and define $\mathbf{X} \in \mathbb{R}^{(2^d-2) \times d}$ by letting the $S$-th row of $\mathbf{X}$ is $\mathbf{1}_S$. Very recently, Zhang et al. (2023a) discovered that

$$\mathbb{E}_{z \in \mathcal{Z}} \left[ \| \phi(z; \boldsymbol{\theta}) - (\mathbf{X}^\top \mathbf{W} \mathbf{X})^{-1} \mathbf{X}^\top \mathbf{W} (\mathbf{U}_z - U_z(\emptyset) \mathbf{1}_{2^d-2}) \|_{\mathbf{X}^\top \mathbf{W} \mathbf{X}}^2 \right]$$

$$= \mathbb{E}_{z \in \mathcal{Z}} \left[ \| \mathbf{U}_z - U_z(\emptyset) \mathbf{1}_{2^d-2} - \mathbf{X} \phi(z; \boldsymbol{\theta}) \|_{\mathbf{W}}^2 + C_z \right]$$

$$= \mathbb{E}_{z \in \mathcal{Z}} \left[ \sum_{\emptyset \subsetneq S \subsetneq [d]} \frac{d-1}{\binom{d}{s} s(d-s)} \left( U_z(S) - U_z(\emptyset) - \mathbf{1}_S^\top \phi(z; \boldsymbol{\theta}) \right)^2 + C_z \right]$$

where $C_z = \| (\mathbf{X}^\top \mathbf{W} \mathbf{X})^{-1} \mathbf{X}^\top \mathbf{W} (\mathbf{U}_z - U_z(\emptyset) \mathbf{1}_{2^d-2}) \|_{\mathbf{X}^\top \mathbf{W} \mathbf{X}}^2 - \| U_z(\emptyset) \mathbf{1}_{2^d-2} - \mathbf{U}_z \|_{\mathbf{W}}^2$.

The convention is $\| \mathbf{z} \|_{\mathbf{A}}^2 := \mathbf{z}^\top \mathbf{A} \mathbf{z}$. This relationship reveals that FastSHAP is just to train explainers towards a (potentially) biased target $(\mathbf{X}^\top \mathbf{W} \mathbf{X})^{-1} \mathbf{X}^\top \mathbf{W} (\mathbf{U}_z - U_z(\emptyset) \mathbf{1}_{2^d-2})$ under the metric induced by $\mathbf{X}^\top \mathbf{W} \mathbf{X}$; see Proposition 9. Therefore, they instead proposed to train explainers based on

$$\mathbb{E}_{z \sim \mathcal{Z}} [\| \phi(z; \theta) - \phi \|^2] \tag{20}$$

where $\phi$ is any unbiased estimator for the Shapley value. They demonstrated in the experiments that this framework of training explainers is not just simple but effective. Note that the target $(\mathbf{X}^\top \mathbf{W} \mathbf{X})^{-1} \mathbf{X}^\top \mathbf{W} (\mathbf{U}_z - U_z(\emptyset) \mathbf{1}_{2^d-2})$ is expected to be biased because compared with Eq. (3), the efficiency constraint, which is supposed to be $\sum_{i=1}^d \phi(z; \boldsymbol{\theta}) = U_z([d]) - U_z(\emptyset)$, has been removed. Nevertheless, to overcome this issue, the authors of FastSHAP added an additive-efficient-normalization layer on top during training and inference (Jethani et al. 2022, Table 3), which is $\phi(z; \boldsymbol{\theta}) \leftarrow \phi(z; \boldsymbol{\theta}) + \frac{1}{d}(U_z([d]) - U_z(\emptyset) - \sum_{i=1}^d \phi_i(z; \boldsymbol{\theta})) \mathbf{1}_d$.

**Remark 6.** *Note that the framework (20) can be used for training explainers towards any probabilistic value. Intuitively, a faster unbiased estimator substituted in the framework (20) would lead to better training of explainers. All in all, faster estimators could possibly benefit the research line of training probabilistic-value-based explainers in feature attribution.*

**Proposition 9.** *Let $\hat{\boldsymbol{\phi}}^{Sh} = (\mathbf{X}^\top \mathbf{W} \mathbf{X})^{-1} \mathbf{X}^\top \mathbf{W} (\mathbf{U}_z - U_z(\emptyset) \mathbf{1}_{2^d-2})$ and $\boldsymbol{\phi}^{Sh}$ be the Shapley value using the utility function $U_z$. Then, there is*

$$\boldsymbol{\phi}^{Sh} = \hat{\boldsymbol{\phi}}^{Sh} + \rho \mathbf{1}_d$$

$$where \ \rho = -\frac{\mathbf{1}_d^\top (\mathbf{X}^\top \mathbf{W} \mathbf{X})^{-1} \mathbf{X}^\top \mathbf{W} (\mathbf{U}_z - U_z(\emptyset) \mathbf{1}_{2^d-2}) - U_z([d]) + U_z(\emptyset)}{\mathbf{1}_d^\top (\mathbf{X}^\top \mathbf{W} \mathbf{X})^{-1} \mathbf{1}_d}.$$

*In other words, $(\mathbf{X}^\top \mathbf{W} \mathbf{X})^{-1} \mathbf{X}^\top \mathbf{W} (\mathbf{U}_z - U_z(\emptyset) \mathbf{1}_{2^d-2})$ is potentially biased as $\rho$ is not necessarily zero.*

*Proof.* In view of the optimization problem (17), let $m_s \leftarrow \frac{d-1}{\binom{d}{s}s(d-s)}$, $n \leftarrow d$ and $U^n \leftarrow U_z$. By (Charnes et al. 1988, Theorem 4), the induced uniquely optimal solution $\mathbf{v}^*$ is exactly $\phi^{Sh}$. According to Eq. (19), we have

$$\mathbf{v}^* = \mathbf{A}^{-1}\left(\mathbf{b} - \frac{\mathbf{1}_d^\top \mathbf{A}^{-1}\mathbf{b} - U_z([d]) + U_z(\emptyset)}{\mathbf{1}_d^\top \mathbf{A}^{-1}\mathbf{1}_d}\mathbf{1}_d\right)$$
$$\text{where } \mathbf{A} = \mathbf{X}^\top \mathbf{W}\mathbf{X} \text{ and } \mathbf{b} = \mathbf{X}^\top \mathbf{W}(\mathbf{U}_z - U_z(\emptyset)\mathbf{1}_{2^d - 2}),$$

from which we can deduce

$$\phi^{Sh} = \mathbf{v}^* = \mathbf{A}^{-1}\mathbf{b} + \rho\mathbf{1}_d = \hat{\phi}^{Sh} + \rho\mathbf{1}_d$$
$$\text{where } \rho = -\frac{\mathbf{1}_d^\top \mathbf{A}^{-1}\mathbf{b} - U_z([d]) + U_z(\emptyset)}{\mathbf{1}_d^\top \mathbf{A}^{-1}\mathbf{1}_d}.$$

$\square$

## H    Interpretation of $\kappa(N)$ and $\tau(N)$ in Proposition 4

Recall that $\kappa(n) = \gamma(n)^{-2}$ where $\gamma(n) = \frac{\sum_{s=1}^{n-1}\binom{n-1}{s-1}(p_s^n + p_{s+1}^n)}{\sum_{s=1}^{n-1}\binom{n}{s}(p_s^n + p_{s+1}^n)}$. Our interpretation of $\kappa(n)$ is based on $\gamma(n)$. In Algorithm 1, for each non-empty proper subset $S$, its probability of being sampled is

$$P(S) = \frac{p_s^n + p_{s+1}^n}{\sum_{s=1}^{n-1}\binom{n}{s}(p_s^n + p_{s+1}^n)}.$$

Since $\gamma(n) = \sum_{S \subsetneq [n]:\, i \in S} P(S)$, $\gamma(n)$ is the probability that the $i$-th data point appears in a random sample $S$. Thanks to symmetry, the choice of $i$ is immaterial. Thus,

$$n\gamma(n) = \sum_{i=1}^{n}\sum_{S \subsetneq [n]:\, i \in S} P(S) = \sum_{S \subsetneq [n]}\sum_{i=1}^{n}\mathbb{1}_{i \in S}P(S) = \sum_{\emptyset \subsetneq S \subsetneq [n]} sP(S) = \mathbb{E}_S[s],$$

whence follows

$$\gamma(n) = \frac{\mathbb{E}_S[s]}{n} \geq \frac{1}{n}.$$

The lower bound is achieved when $p_1^n = 1$ and $p_i^n = 0$ for all $i \geq 2$, corresponding to the semivalue with $\mu = \delta_0$, i.e., leave everything else out.

Looking into Algorithm 1, for each sampled utility evaluation $U^n(S)$, it is used to update the estimates of $s$ data. Therefore, $\gamma(n)$ is just the average rate of reusing utility evaluations, and $\kappa(n)$ is the inverse square of this average rate. The higher $\gamma(n)$ is, the more efficient GELS-R is. On the flip side, reusing utility evaluations also creates correlation among the estimates. We conclude that for the benefit to outweigh the cost, $\gamma(n)$ needs to be on the order of $\omega(\frac{1}{\sqrt{n}})$, which is easily satisfied as long as the sequence $\{p_s^n\}$ is not exclusively concentrated around small $s$. All practical probabilistic values used in the literature, including the ones in our experiments, meet this condition.

Recall that $\tau(n) = \left(\sum_{s=1}^{n}\binom{n+1}{s}p_s^n\right)^2$ and we write $\zeta(n) = \tau(n)^{-\frac{1}{2}}$. Observe that, for every $i \in [n]$,

$$2\zeta(n) = \lambda \cdot \frac{1}{\sum_{s=1}^{n}\binom{n}{s}p_s^n} + (1-\lambda) \cdot \frac{1}{\sum_{s=1}^{n}\binom{n}{s-1}p_s^n}$$

where $\lambda = \frac{\sum_{s=1}^{n}\binom{n}{s}p_s^n}{\sum_{s=1}^{n}\binom{n+1}{s}p_s^n}$ is the probability of subsets sampled by Algorithm 2 not containing the introduced null data point (equivalently, the $(n+1)$-th data point). Then, for any $i \in [n]$,

$$\frac{1}{\sum_{s=1}^{n}\binom{n}{s}p_s^n} = \frac{\sum_{s=1}^{n}\binom{n-1}{s-1}p_s^n}{\sum_{s=1}^{n}\binom{n}{s}p_s^n} = \sum_{\emptyset \subsetneq S \subseteq [n]:\, i \in S} P_{\not\ni n+1}(S)$$

where $P_{\not\ni n+1}(S)$ is the probability of the set $S$ sampled by Algorithm 2 conditioned on that the samples do not contain the $(n+1)$-th data point. Therefore,

$$\frac{n}{\sum_{s=1}^{n} \binom{n}{s} p_s^n} = \sum_{i=1}^{n} \sum_{\emptyset \subsetneq S \subseteq [n]} \mathbb{1}_{i \in S} P_{\not\ni n+1}(S) = \sum_{\emptyset \subsetneq S \subseteq [n]} s P_{\not\ni n+1}(S) = \mathbb{E}_{S|S \not\ni n+1}[s].$$

On the other hand, for every $i \in [n]$,

$$\frac{1}{\sum_{s=1}^{n} \binom{n}{s-1} p_s^n} = \frac{\sum_{s=1}^{n} \binom{n-1}{s-1} p_s^n}{\sum_{s=1}^{n} \binom{n}{s-1} p_s^n} = \sum_{\emptyset \subsetneq S \subsetneq [n+1]:\ i \notin S, n+1 \in S} P_{\ni n+1}(S),$$

which leads to

$$\frac{n}{\sum_{s=1}^{n} \binom{n}{s-1} p_s^n} = \sum_{i=1}^{n} \sum_{\emptyset \subsetneq S \subsetneq [n+1]:\ n+1 \in S} \mathbb{1}_{i \notin S} P_{\ni n+1}(S)$$

$$= \sum_{\emptyset \subsetneq S \subsetneq [n+1]:\ n+1 \in S} (n+1-s) P_{\ni n+1}(S) = \mathbb{E}_{S|S \ni n+1}[n+1-s].$$

Eventually, we have

$$\zeta(n) = \frac{1}{2n} \left( \lambda \cdot \mathbb{E}_{S|S \not\ni n+1}[s] + (1-\lambda) \cdot \mathbb{E}_{S|S \ni n+1}[n+1-s] \right)$$

$$= \frac{1}{2n} \mathbb{E}_S[s \mathbb{1}_{n+1 \notin S} + (n+1-s)\mathbb{1}_{n+1 \in S}].$$

Looking into Algorithm 2, i) if the sampled subset $S$ does not contain the null data, there are $s$ of $\{\hat{r}_i - \hat{r}_{n+1}\}_{1 \leq i \leq n}$ receiving updates from $U^n(S)$; ii) for the other way, there are $n+1-s$ of them receiving updates from $U^n(S)$. To conclude, $\frac{\tau(n)}{4}$ is the square inverse of the average rate of reusing utility evaluations while running Algorithm 2.

## I ASYMPTOTIC ANALYSIS

This appendix is mainly to analyze the asymptotic behavior (as $n \to \infty$) of $\kappa(n)$ and $\tau(n)$ that appear in Proposition 4. For probabilistic values, generally, there is no restriction on how $\{\mathbf{p}^n \in \mathbb{R}^n\}_{n \geq 1}$ are organized. Therefore, to analyze the convergence of our proposed estimators, we focus on semi-values instead. According to Dubey et al. (1981), each semi-value corresponds to a probability measure $\mu$ on the interval $[0, 1]$ such that

$$p_s^n = \int_0^1 t^{s-1}(1-t)^{n-s} \mathrm{d}\mu(t) \text{ for every } 1 \leq s \leq n. \tag{21}$$

**Lemma 2.** *Suppose $n > 1$, and for each probability measure $\mu$ on the closed interval $[0,1]$, define*

$$m_k^\mu = \int_0^1 t^k \mathrm{d}\mu(t) \text{ and } w_k^\mu = \int_0^1 (1-t)^k \mathrm{d}\mu(t) \text{ for every } k \geq 0,$$

$$M_k^\mu = \sum_{j=0}^{k} m_k^\mu \text{ and } W_k^\mu = \sum_{j=0}^{k} w_k^\mu \text{ for every } k \geq 0.$$

*Then, there is*

$$\kappa(n)^{\frac{1}{2}} = \frac{M_{n-2}^\mu + W_{n-2}^\mu}{M_{n-2}^\mu} \text{ and } \tau(n)^{\frac{1}{2}} = M_{n-1}^\mu + W_{n-1}^\mu.$$

*Proof.* Recall that in Proposition 4 $\kappa(n)^{\frac{1}{2}} = \frac{\sum_{s=1}^{n-1} \binom{n}{s}(p_s^n + p_{s+1}^n)}{\sum_{s=1}^{n-1} \binom{n-1}{s-1}(p_s^n + p_{s+1}^n)}$. By Eq. (21), there is $p_s^n + p_{s+1}^n = p_s^{n-1}$ for every $1 \leq s \leq n-1$. Notice that

$$\sum_{s=1}^{n-1} \binom{n}{s} p_s^{n-1} = \sum_{s=1}^{n-1} \binom{n-1}{s} p_s^{n-1} + \sum_{s=1}^{n-1} \binom{n-1}{s-1} p_s^{n-1}.$$

By Eq. (21),

$$\sum_{s=1}^{n-1} \binom{n-1}{s} p_s^{n-1} = \int_0^1 \sum_{s=1}^{n-1} \binom{n-1}{s} t^{s-1}(1-t)^{n-1-s} \mathrm{d}\mu(t)$$

$$= \int_0^1 \frac{1}{t} \sum_{s=1}^{n-1} \binom{n-1}{s} t^s (1-t)^{n-1-s} \mathrm{d}\mu(t)$$

$$= \int_0^1 \frac{1-(1-t)^{n-1}}{t} \mathrm{d}\mu(t) = \int_0^1 \frac{(1-(1-t))(\sum_{j=0}^{n-2}(1-t)^j)}{t} \mathrm{d}\mu(t)$$

$$= W_{n-2}^\mu$$

Note $\sum_{s=1}^{n-1} \binom{n-1}{s} t^{s-1}(1-t)^{n-1-s} = \sum_{j=0}^{n-2}(1-t)^j$ still holds for $t = 0$. Similarly, one can get

$$\sum_{s=1}^{n-1} \binom{n-1}{s-1} p_s^{n-1} = M_{n-2}^\mu.$$

Recall that in Proposition 4 $\tau(n)^{\frac{1}{2}} = \sum_{s=1}^{n} \binom{n+1}{s} p_s^n$, and thus one can get $\tau(n)^{\frac{1}{2}} = M_{n-1}^\mu + W_{n-1}^\mu$ in a similar fashion.
$\square$

Using the monotone convergence theorem, we have

$$\lim_{n\to\infty} M_n^\mu = \lim_{n\to\infty} \int_0^1 \frac{1-t^{n+1}}{1-t} \mathrm{d}\mu(t) = \int_0^1 \lim_{n\to\infty} \frac{1-t^{n+1}}{1-t} \mathrm{d}\mu(t) = \int_0^1 \frac{1}{1-t} \mathrm{d}\mu(t),$$

$$\lim_{n\to\infty} W_n^\mu = \lim_{n\to\infty} \int_0^1 \frac{1-(1-t)^{n+1}}{t} \mathrm{d}\mu(t) = \int_0^1 \frac{1}{t} \mathrm{d}\mu(t). \tag{22}$$

Note that the extreme cases, e.g., $\lim_{n\to\infty} M_n^{\delta_1} = \int_0^1 \frac{1}{1-t} \mathrm{d}\delta_1(t)$ is trivially true where the convention is $\frac{1}{0} = \infty$. Therefore, $\tau(n) \in \Theta(1)$ if and only if the two integrals in the above are finite. In particular, it holds if the probability density function $p$ of $\mu$ satisfies $\lim_{t\to 0} \frac{p(t)}{t^a} = \lim_{t\to 1} \frac{p(t)}{(1-t)^b} = 0$ for some $a, b > 0$; see Proposition 10. Interestingly, we have

$$\kappa(n)^{\frac{1}{2}} = 1 + \frac{W_{n-2}^\mu}{M_{n-2}^\mu} \le 1 + W_{n-2}^\mu.$$

Thus, $\kappa(n) = \Theta(1)$ if the integral Eq. (22) is finite, meaning that $\mu$ cannot put "large" mass around $t = 0$. However, this is not necessary as any symmetric probability measure $\mu$, e.g., the uniform distribution that leads to the Shapley value, will have $\kappa(n) = 4$.

**Remark 7.** *For extreme cases $\mu = \delta_0$ (leave everything else out) and $\mu = \delta_1$ (leave one out), it is clear that $\kappa(n) \in \Theta(1)$ for the latter but it becomes $\Theta(n^2)$ for the former, in which case our GELS-R (Algorithm 1) in fact only samples subsets of size 1, and therefore each utility evaluation is used to update the estimate of one data point. By contrast, the (weighted) sampling lift estimator always spends two utility evaluations to update the estimate of one data point.*

**Corollary 1.** *For the Banzhaf value, $\kappa(n), \tau(n) \in \Theta(1)$. In other words, GELS, GELS-R and GELS-Shapley require $O(\frac{n}{\epsilon^2} \log \frac{n}{\delta})$ utility evaluations to achieve an $(\epsilon, \delta)$-approximation for the Banzhaf value.*

*Proof.* For the Banzhaf value, $\mu = \delta_{0.5}$ (Dirac delta distribution), and thus $\lim_{n\to\infty} M_n^{\delta_{0.5}} = \lim_{n\to\infty} W_n^{\delta_{0.5}} = 2$.
$\square$

We now provide some easily verifiable conditions that determine the growth of $\kappa(n)$ and $\tau(n)$. Recall that as long as $\kappa(n), \tau(n) = o(n)$, our estimators are more efficient than the sampling lift estimator.

**Proposition 10.** *Assume the probability measure $\mu$ admits a density function $p$ such that $\mu(S) = \int_S p(t) \mathrm{d}t$ for every Borel-measurable subset $S \subseteq [0, 1]$, then,*

1. $\kappa(n), \tau(n) \in \Theta(1)$ *if there exist* $a, b > 0$ *such that* $\lim_{t \to 0} \frac{p(t)}{t^a} = \lim_{t \to 1} \frac{p(t)}{(1-t)^b} = 0.$

2. $\kappa(n), \tau(n) \in O(\log(n)^2)$ *if* $\limsup_{t \to 0} p(t) < \infty$ *and* $\limsup_{t \to 1} p(t) < \infty$. *Examples include if $p$ is bounded (the so-called continuous semivalues used by Dubey et al. (1981)).*

*Proof.* We first show that there exist counterexamples if these conditions are violated. For the Shapley value with $\mu = \mathcal{U}$ (the uniform measure), $m_k^{\mathcal{U}} = w_k^{\mathcal{U}} = \frac{1}{k+1}$, and thus $\tau(n) = \Theta(\log(n)^2)$. For the second, looking into Lemma 3, consider $\alpha = \beta < 1$, $\tau(n) \in \Theta(n^{2-2\alpha})$.

Suppose there exist $a, b > 0$ such that $\lim_{t \to 0} \frac{p(t)}{t^a} = \lim_{t \to 1} \frac{p(t)}{(1-t)^b} = 0$. Then, there exists some $\epsilon, C > 0$ such that $p(t) \le Ct^a(1-t)^b$ if $t < \epsilon$ and $t > 1 - \epsilon$. Define a positive measure $\mu_\epsilon$ by letting $\mu_\epsilon(S) = \mu(S \cap [\epsilon, 1 - \epsilon])$ for every Borel-measurable subset $S \subseteq [0, 1]$. Besides, define another positive measure $\lambda$ by letting $\lambda(S) = \int_S Ct^a(1-t)^b dt$ for every Borel-measurable subset $S \subseteq [0, 1]$. Therefore, we have $\mu \le \mu_\epsilon + \lambda$, which indicates that $m_k^\mu \le m_k^{\mu_\epsilon} + m_k^\lambda$, and thus $M_k^\mu \le M_k^{\mu_\epsilon} + M_k^\lambda$. By Lemma 3, $M_k^\lambda \in \Theta(1)$. For the other, observe that $m_k^{\mu_\epsilon} \le m_k^{\delta_{1-\epsilon}}$, and therefore $M_k^{\mu_\epsilon} \le M_k^{\delta_{1-\epsilon}} \in \Theta(1)$. The remaining case $W_k^\mu$ can be tackled similarly. To conclude, $\kappa(n), \tau(n) \in \Theta(1)$.

Suppose $\limsup_{t \to 0} p(t) < \infty$ and $\limsup_{t \to 1} p(t) < \infty$, then there exist $\epsilon, C > 0$ such that $p(t) \le C$ for every $t \in [0, \epsilon) \cup (1 - \epsilon, 1]$. Define $\mu_\epsilon$ as the one in the above. Then, $\mu \le \mu_\epsilon + C\mathcal{U}$ where $\mathcal{U}$ denotes the uniform measure. Therefore, $w_k^\mu \le w_k^{\mu_\epsilon} + w_k^{C\mathcal{U}}$. Since $w_k^{C\mathcal{U}} = \frac{C}{k+1}$, there is $W_k^{C\mathcal{U}} \in \Theta(\log n)$. Besides, $W_k^{\mu_\epsilon} \le W_k^{\delta_\epsilon} \in \Theta(1)$. The remaining case $M_k^\mu$ can be dealt with similarly. Therefore, the conclusion follows by using Lemma 2. $\square$

Next, we derive a more precise estimate for the Beta Shapley values.

**Lemma 3.** *Let $B(\alpha, \beta)$ be the beta distribution with probability density function $\propto t^{\alpha-1}(1-t)^{\beta-1}$. Note that if $\mu = B(\alpha, \beta)$, it yields the Beta$(\beta, \alpha)$ (a parameterized Beta Shapley value).*

$$M_n^{B(\alpha,\beta)} \in \begin{cases} \Theta(\log n) & \beta = 1 \\ \Theta(1) & \beta > 1 \\ \Theta(n^{1-\beta}) & 0 < \beta < 1 \end{cases} \quad and \quad W_n^{B(\alpha,\beta)} \in \begin{cases} \Theta(\log n) & \alpha = 1 \\ \Theta(1) & \alpha > 1 \\ \Theta(n^{1-\alpha}) & 0 < \alpha < 1 \end{cases}.$$

*Proof.* Let $\Gamma$ denote the Gamma function.

$$m_k^{B(\alpha,\beta)} = \frac{\Gamma(\alpha+\beta)}{\Gamma(\alpha)\Gamma(\beta)} \cdot \frac{\Gamma(\alpha+k)\Gamma(\beta)}{\Gamma(\alpha+\beta+k)} = \frac{\prod_{j=0}^{k-1}(\alpha+j)}{\prod_{j=0}^{k-1}(\alpha+\beta+j)}.$$

We write $a_k \sim b_k$ if $\lim_{k \to \infty} \frac{a_k}{b_k}$ converges. Since $\Gamma(x) = \lim_{n \to \infty} \frac{n!n^x}{x(x+1)\cdots(x+n)}$, e.g., see (Rudin 1953, Eq. (95) in Chapter 8), we have $m_k^{B(\alpha,\beta)} \sim \frac{k!k^\alpha}{k!k^{\alpha+\beta}} = k^{-\beta}$, and thus the conclusion follows. The remaining case can be derived similarly. $\square$

**Remark 8.** *Lemma 3 is useful for obtaining the time complexity of the proposed estimators for the Beta Shapley values parameterized by $\alpha, \beta > 0$. If $\alpha, \beta > 1$, the two proposed estimators require $O(\frac{n}{\epsilon^2} \log \frac{n}{\delta})$ utility evaluations to achieve an $(\epsilon, \delta)$-approximation. Note that this is the currently best time complexity. For the Shapley value, i.e., Beta$(1, 1)$, it is $O(\frac{n}{\epsilon^2} \log \frac{n}{\delta})$ for GELS-R, and $O(\frac{n}{\epsilon^2} \log(\frac{n}{\delta}) \log(n)^2)$ for GELS and GELS-Shapley. To our knowledge, in terms of $(\epsilon, \delta)$-approximation, the previously best time complexity for the Shapley value is $O(\frac{n}{\epsilon^2} \log(\frac{n}{\delta}) \log n)$ achieved by the group testing estimator (Wang and Jia 2023, Theorem C.7).*

## J   MORE EXPERIMENT RESULTS

For training estimators on MNIST, all performance curves are provided in Figure 11. It can be seen that the conclusions we have in the main paper still hold on MNIST.

The paired sampling technique was proposed by Covert and Lee (2021) to enhance (unbiased) KernelSHAP, but we notice that it can also be employed for GELS, GELS-R, GELS-Shapley, (weighted)

sampling lift, group testing, AME, MSR and simSHAP. Precisely, suppose the sampled subsets is $\{S_k\}_{k \geq 1}$, the paired sampling employs sampled subsets $\{T_j\}_{j \geq 1}$ such that $T_{2k-1} = S_k$ and $T_{2k} = [n] \backslash S_k$ for every $k \geq 1$. Roughly speaking, symmetric probabilistic values, i.e., $p_s^n = p_{n-s}^n$ for every $s \in [n]$ could possibly take advantage of this technique. Examples include the Shapley value, the Banzhaf value and Beta$(\gamma, \gamma)$. An exception is that weighted sampling lift can be coupled with the paired sampling for any probabilistic value. In this appendix, we implement the paired sampling technique if possible. The results with utility functions reporting the classification accuracy on $D_{perf}$ are shown in Figures 3, 4, 5 and 6. Moreover, we also set the utility functions to report the cross-entropy loss instead, and the corresponding results are presented in Figures 7, 8, 9 and 10.

**Remark 9.** *Observe that AME, simSHAP and group testing gain significant performance boosts using the paired sampling technique, while (unbiased) KernelSHAP takes advantage of this technique occasionally. For other estimators, the paired sampling does not play a noticeable role. Interestingly, while using the paired sampling technique, group testing is exactly equal to GELS, whereas GELS-Shapley, unbiased KernelSHAP and simSHAP are all equal.*

To see the equality between GELS and group testing while using the paired sampling technique, notice that they share the same way of sampling subsets. Therefore, suppose we have one pair of samples $(S_1, S_2)$ where $S_2 = [n+1] \backslash S_1$. According to Algorithm 2, the corresponding $i$-th estimate of GELS is

$$\hat{\phi}_i^{\text{GELS}} = \begin{cases} H_n \cdot (U^n(S_1 \cap [n]) - U^n(S_2 \cap [n])), & i \in S_1 \text{ and } n+1 \notin S_1 \\ H_n \cdot (U^n(S_2 \cap [n]) - U^n(S_1 \cap [n])), & i \notin S_1 \text{ and } n+1 \in S_1 \\ 0, & \text{otherwise} \end{cases}$$

where $H_n = \sum_{s=1}^{n} \frac{1}{s}$. Looking into the procedure of including a dummy player for group testing (Wang and Jia 2023, Appendix C.3), the reader can verify that group testing produces the same $i$-th estimate using the paired samples $(S, [n+1] \backslash S)$.

For the remaining equality while employing the paired sampling technique, observe that they also have the same way of sampling subsets. Again, suppose we have one pair of samples $(S_1, S_2)$ where $S_2 = [n] \backslash S_1$. By Algorithm 3, the corresponding $i$-th estimate of GELS-Shapley is

$$\hat{\phi}_i^{\text{GELS-Shapley}} = \begin{cases} \dfrac{U^n([n]) - U^n(\emptyset)}{n} + \dfrac{H_{n-1} \cdot s_2}{n} (U^n(S_1) - U^n(S_2)), & i \in S_1 \\ \dfrac{U^n([n]) - U^n(\emptyset)}{n} + \dfrac{H_{n-1} \cdot s_1}{n} (U^n(S_2) - U^n(S_1)), & i \in S_2 \end{cases}$$

where $H_{n-1} = \sum_{s=1}^{n-1} \frac{1}{s}$. As proved by Fumagalli et al. (2023, Theorem 4.5), the corresponding $i$-th estimate of unbiased KernelSHAP is

$$\hat{\phi}_i^{\text{unbiased KernelSHAP}} = \frac{\hat{U}^n([n]) - \hat{U}^n(\emptyset)}{n} + \frac{2H_{n-1}}{T} \sum_{t=1}^{T} \hat{U}^n(S_t) \cdot \left( \mathbb{1}_{i \in S_t} - \frac{s_t}{n} \right)$$

where $\hat{U}^n(S) = U^n(S) - U^n(\emptyset)$. On the other hand, the corresponding $i$-th estimate of simSHAP can be expressed as

$$\hat{\phi}_i^{\text{simSHAP}} = \frac{U^n([n]) - U^n(\emptyset)}{n} + \frac{2H_{n-1}}{T} \sum_{t=1}^{T} U^n(S_t) \cdot \left( \mathbb{1}_{i \in S_t} - \frac{s_t}{n} \right). \tag{23}$$

The reader can verify that $\hat{\phi}_i^{\text{GELS-Shapley}} = \hat{\phi}_i^{\text{unbiased KernelSHAP}} = \hat{\phi}_i^{\text{simSHAP}}$ using $T = 2$ and $S_2 = [n] \backslash S_1$. We point out that Eq. (23) is not the original formula of simSHAP but an equivalent one that has been implicitly mentioned in (Fumagalli et al. 2023, Remark B.1 and Appendix B.4) where they argued that the choice of $\hat{U}^n$ is better than $U^n$. Our experiments confirm that unbiased KernelSHAP converges significantly faster than simSHAP while not using the paired sampling technique.

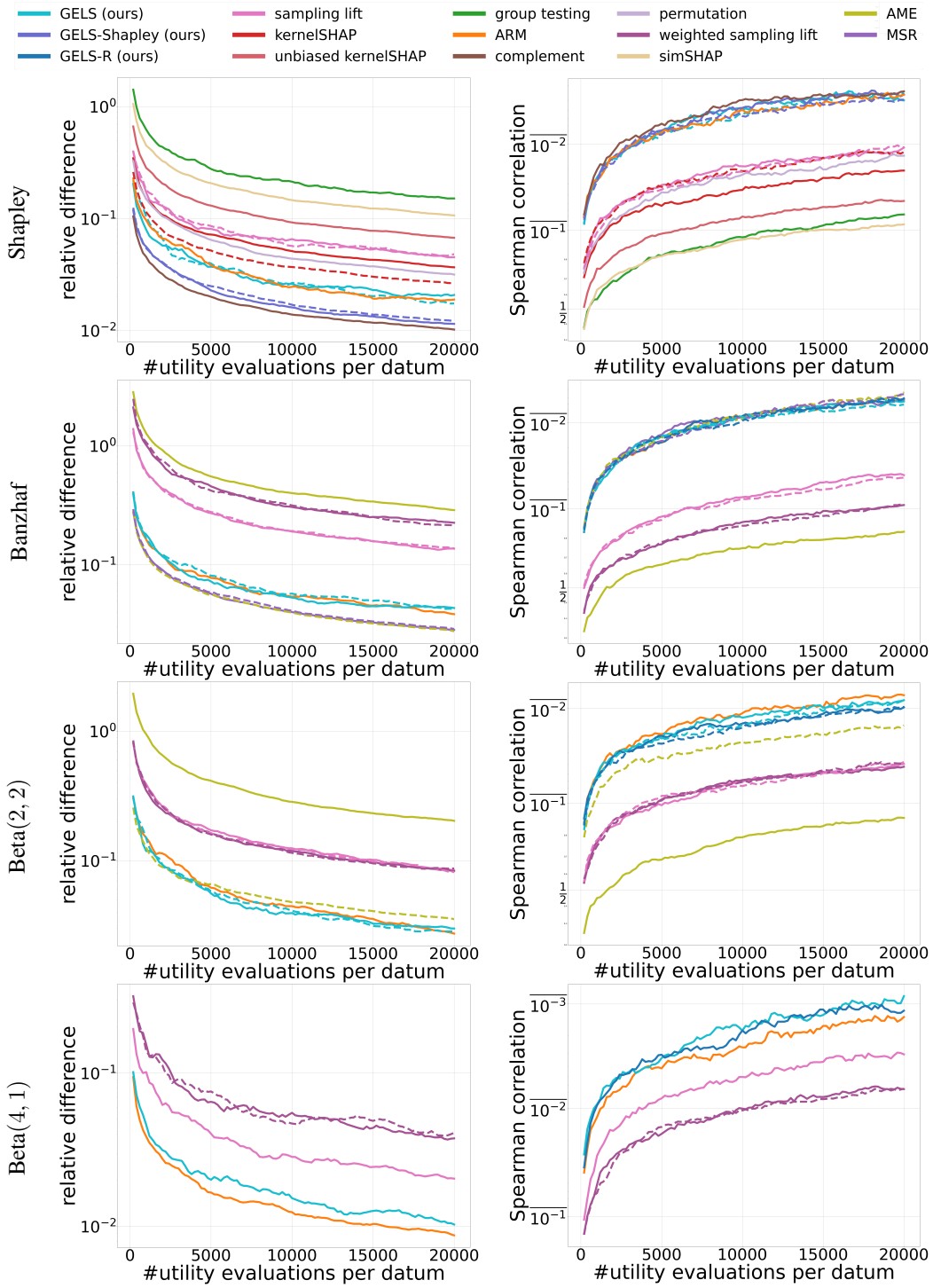

Figure 3: Comparison of different estimators on four probabilistic values using the dataset wind. The relative difference is plotted in log scale while the Spearman correlation is in logit scale. In addition, $\bar{r} = 1 - r$. The utility functions report the classification accuracy on $D_{perf}$. The dashed lines correspond to the use of the paired sampling technique.

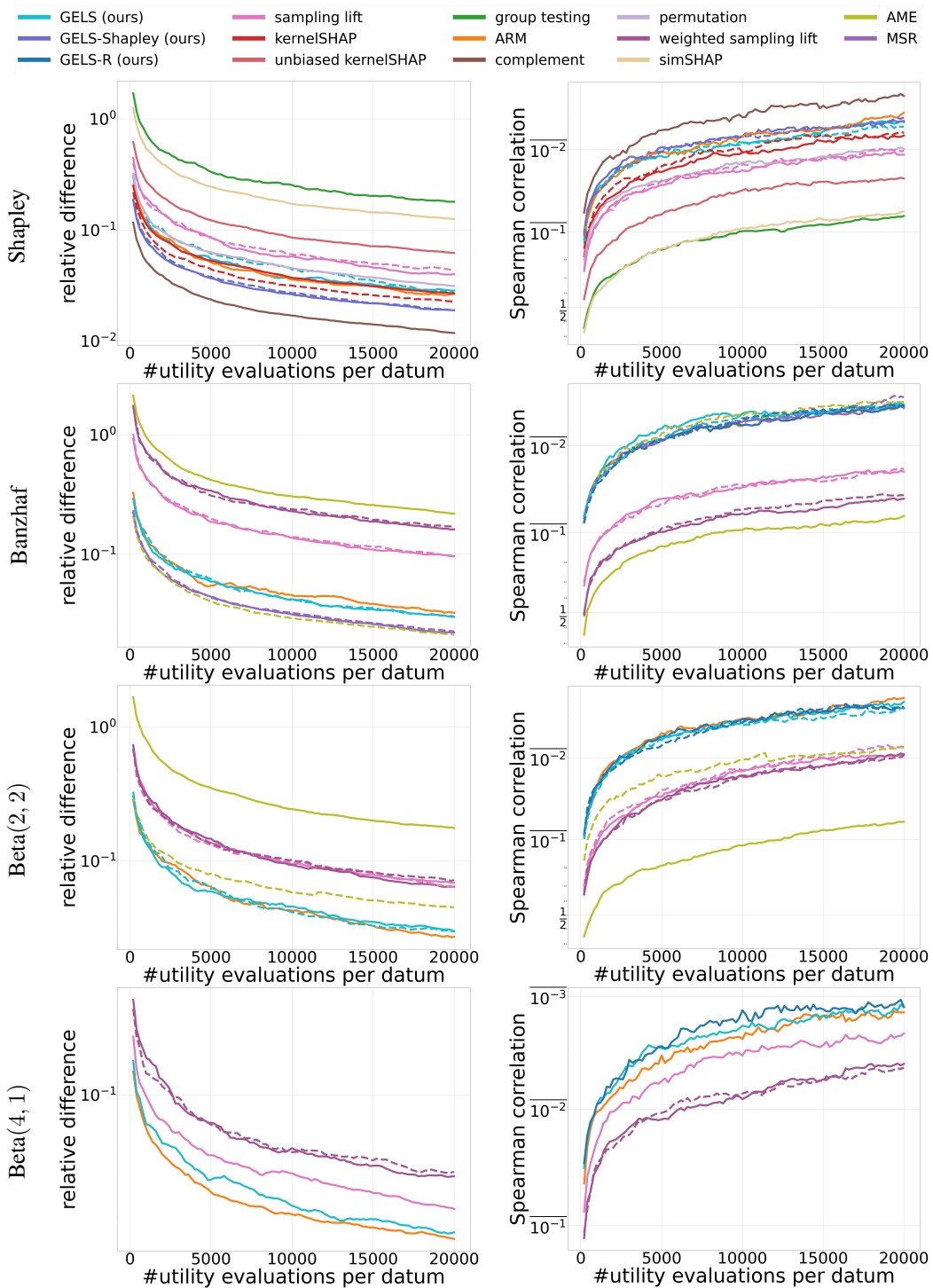

Figure 4: Comparison of different estimators on four probabilistic values using the dataset iris. The relative difference is plotted in log scale while the Spearman correlation is in logit scale. In addition, $\overline{r} = 1 - r$. The utility functions report the classification accuracy on $D_{perf}$. The dashed lines correspond to the use of the paired sampling technique.

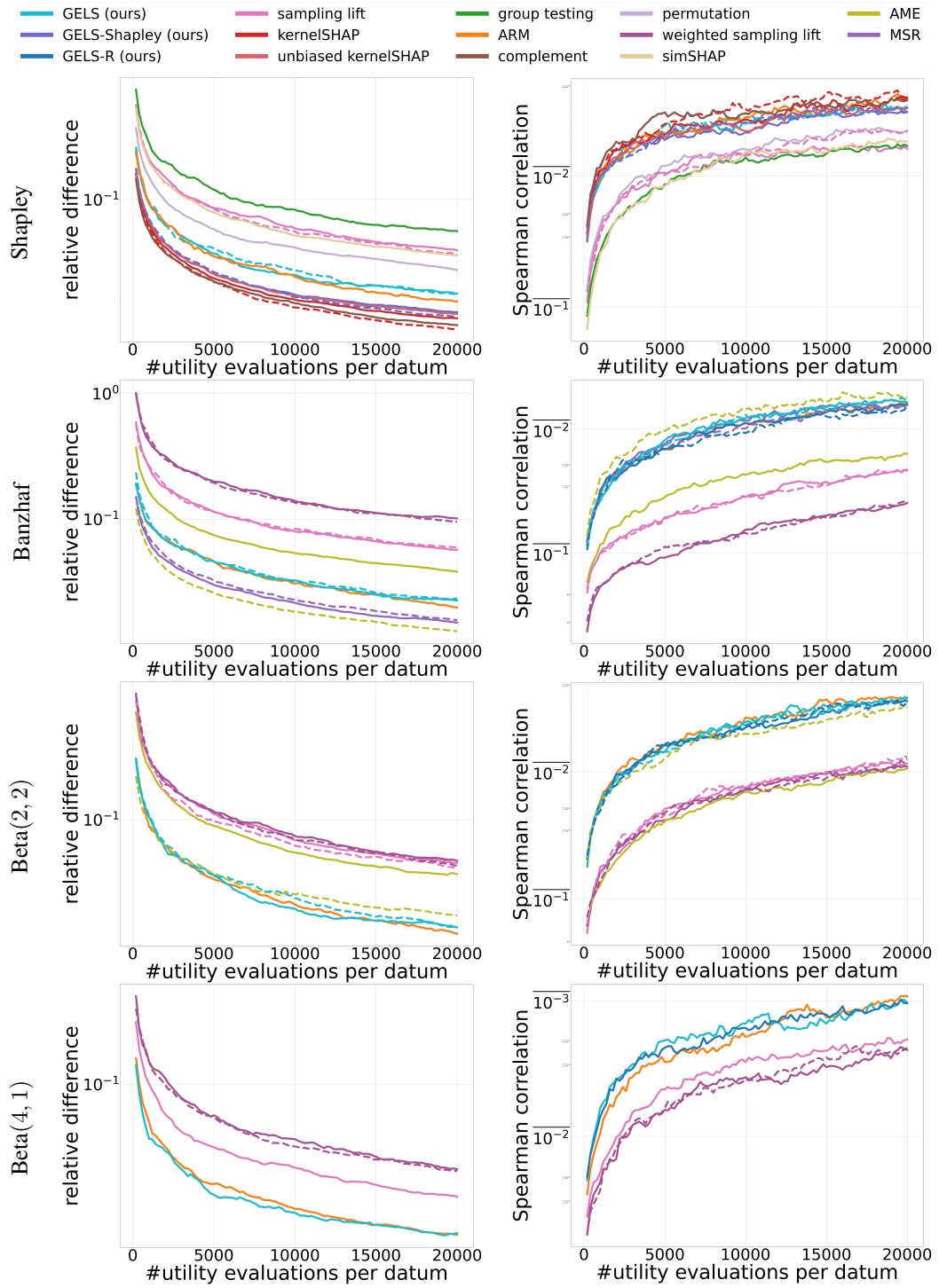

Figure 5: Comparison of different estimators on four probabilistic values using the dataset MNIST. The relative difference is plotted in log scale while the Spearman correlation is in logit scale. In addition, $\overline{r} = 1 - r$. The utility functions report the classification accuracy on $D_{perf}$. The dashed lines correspond to the use of the paired sampling technique.

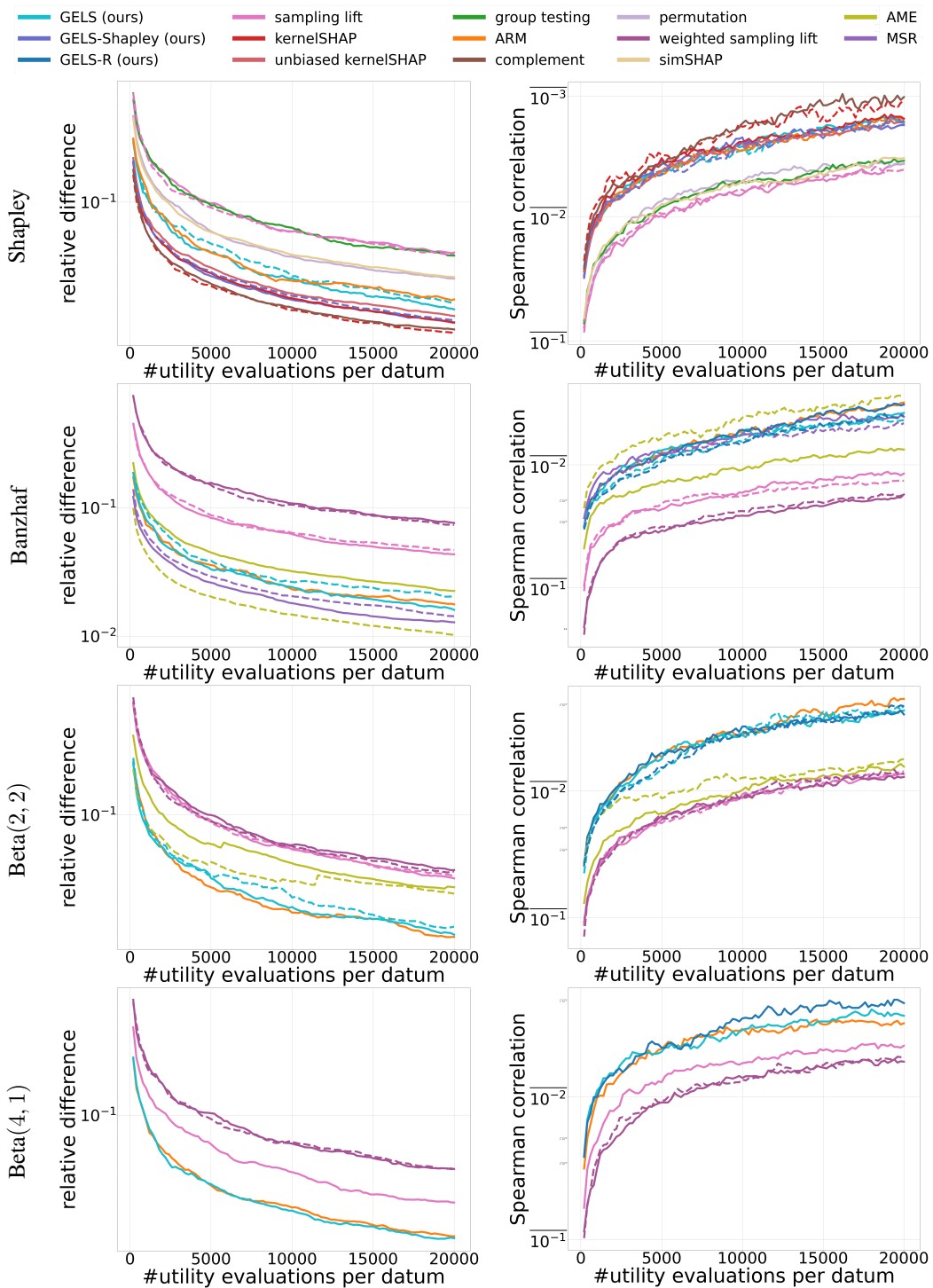

Figure 6: Comparison of different estimators on four probabilistic values using the dataset FMNIST. The relative difference is plotted in log scale while the Spearman correlation is in logit scale. In addition, $\overline{r} = 1 - r$. The utility functions report the classification accuracy on $D_{perf}$. The dashed lines correspond to the use of the paired sampling technique.

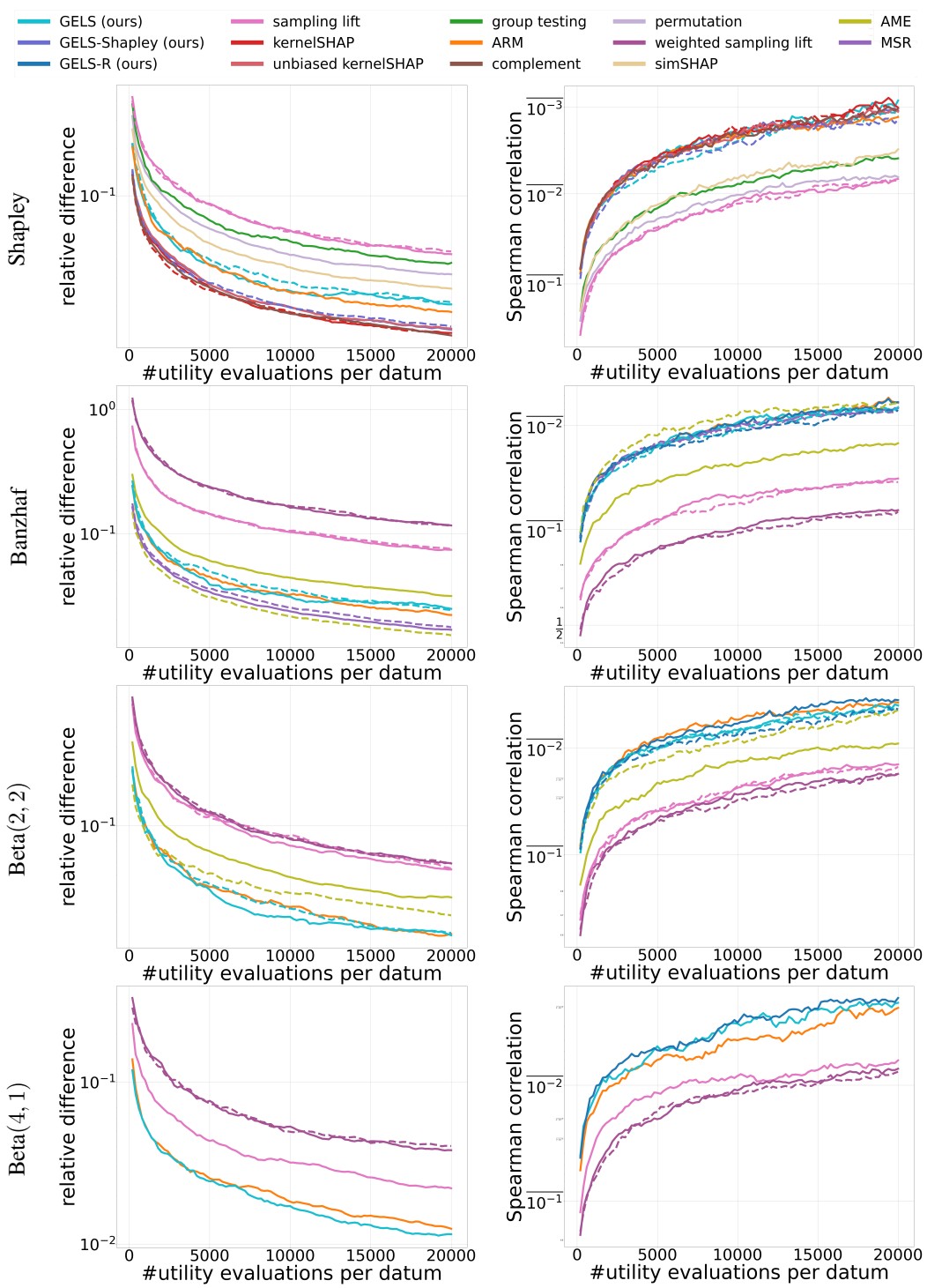

Figure 7: Comparison of different estimators on four probabilistic values using the dataset wind. The relative difference is plotted in log scale while the Spearman correlation is in logit scale. In addition, $\bar{r} = 1 - r$. The utility functions report the cross-entropy loss on $D_{perf}$. The dashed lines correspond to the use of the paired sampling technique.

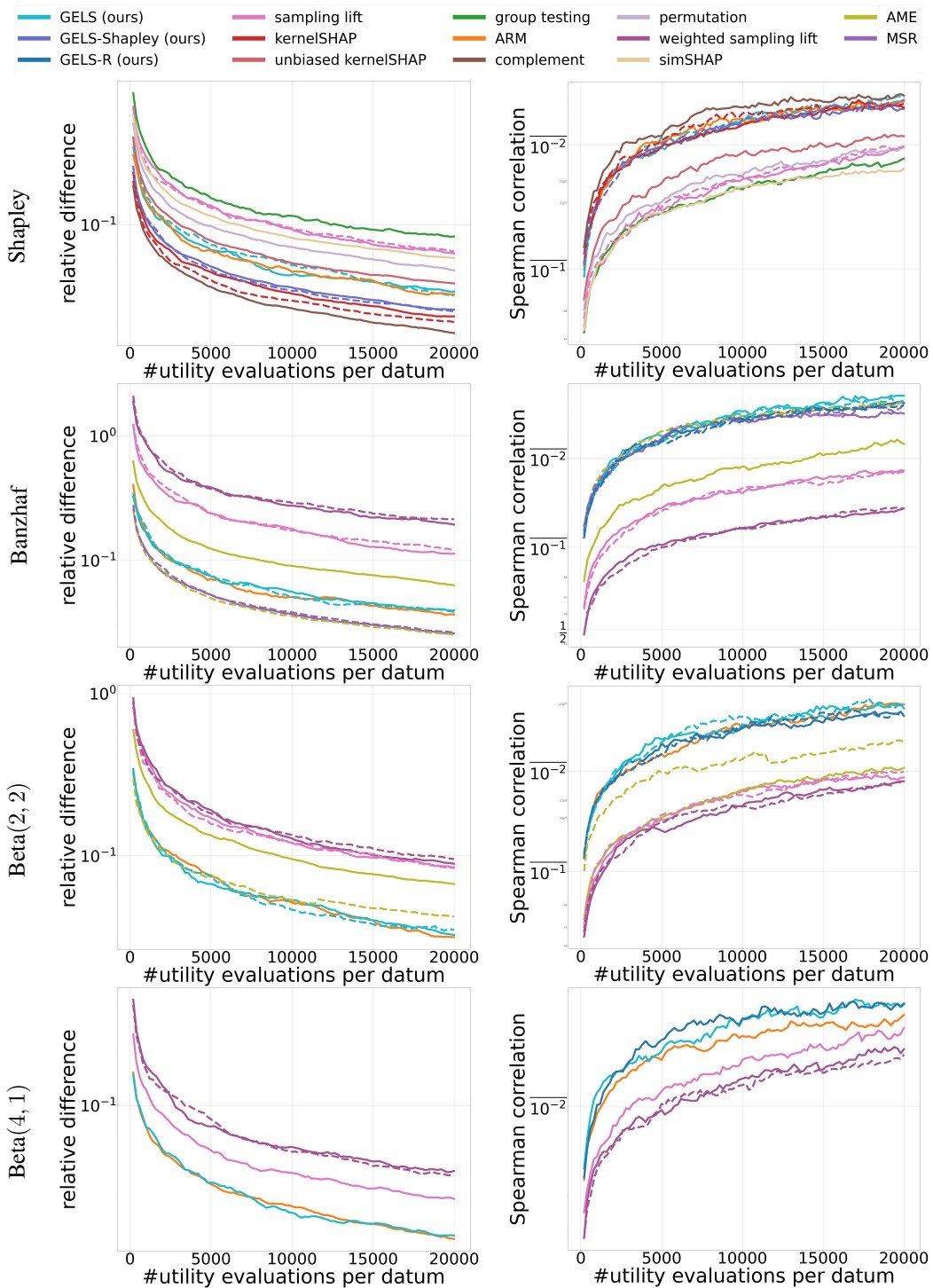

Figure 8: Comparison of different estimators on four probabilistic values using the dataset iris. The relative difference is plotted in log scale while the Spearman correlation is in logit scale. In addition, $\overline{r} = 1 - r$. The utility functions report the cross-entropy loss on $D_{perf}$. The dashed lines correspond to the use of the paired sampling technique.

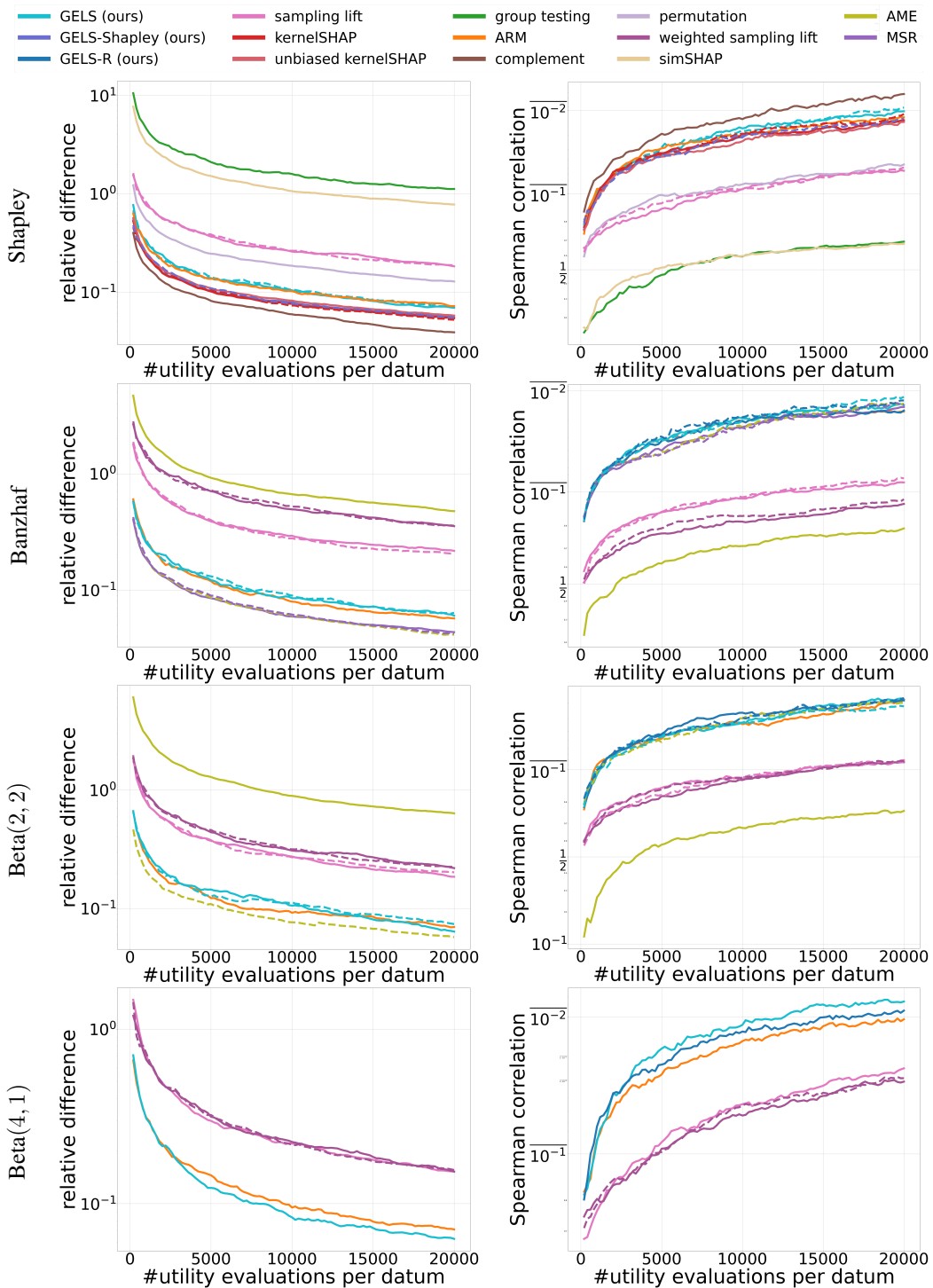

Figure 9: Comparison of different estimators on four probabilistic values using the dataset MNIST. The relative difference is plotted in log scale while the Spearman correlation is in logit scale. In addition, $\overline{r} = 1 - r$. The utility functions report the cross-entropy loss on $D_{perf}$. The dashed lines correspond to the use of the paired sampling technique.

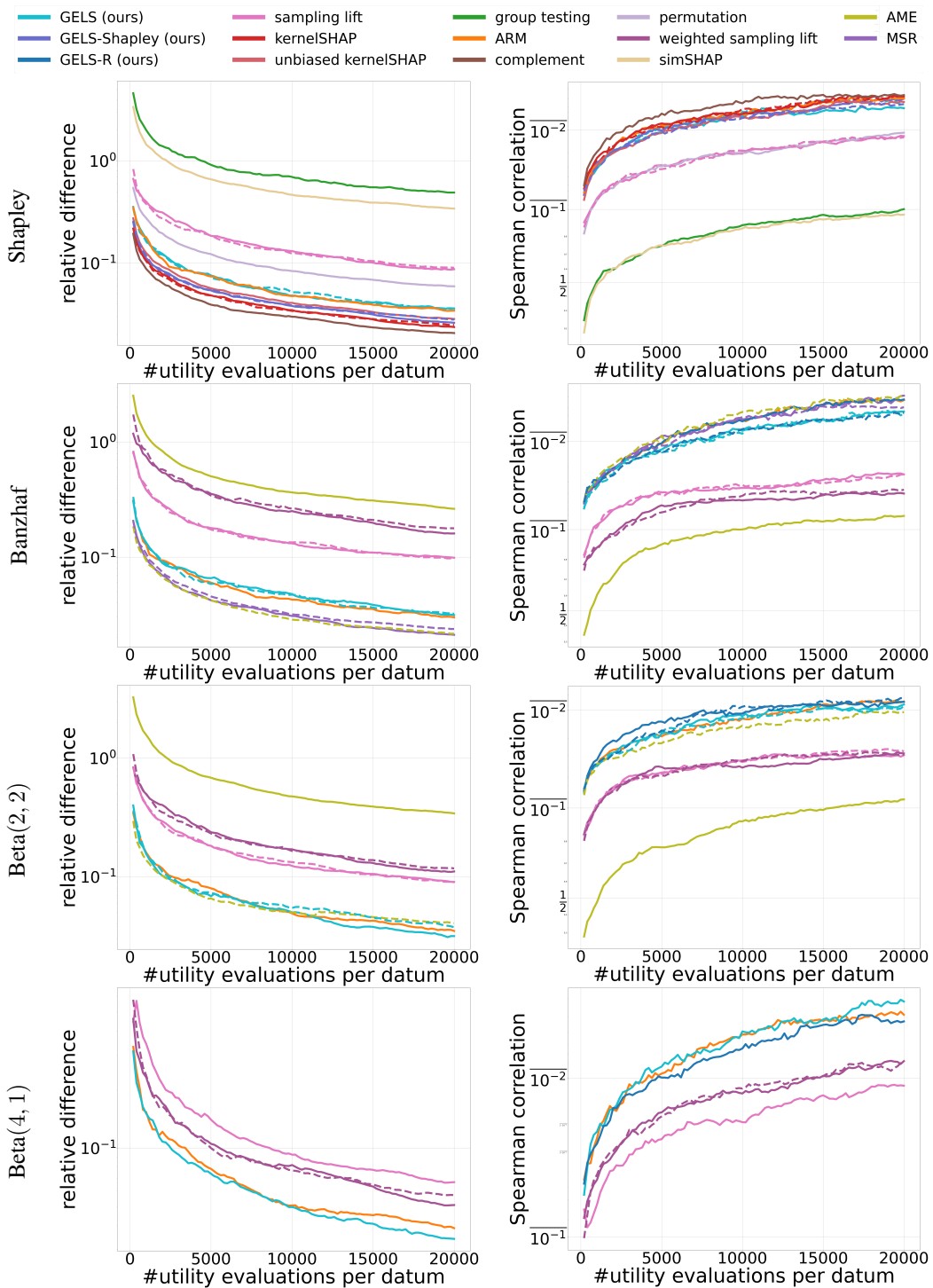

Figure 10: Comparison of different estimators on four probabilistic values using the dataset FM-NIST. The relative difference is plotted in log scale while the Spearman correlation is in logit scale. In addition, $\overline{r} = 1 - r$. The utility functions report the cross-entropy loss on $D_{perf}$. The dashed lines correspond to the use of the paired sampling technique.

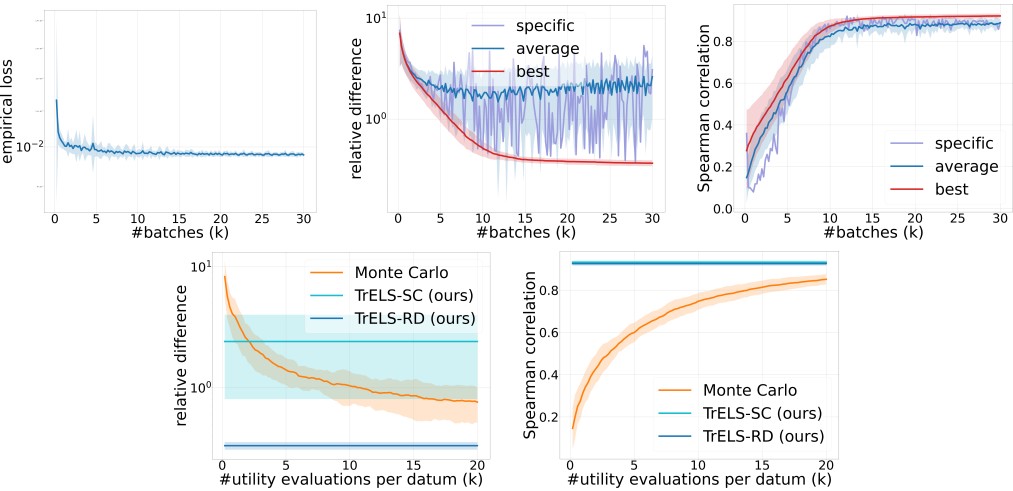

Figure 11: i) The first row: The first plot is the empirical loss of the sampled batch, while the others present the relative distance $\|\hat{\varphi} - \varphi\|_2 / \|\varphi\|_2$ and the Spearman correlation between $\varphi$ and $\hat{\varphi}$. $\hat{\varphi}$ denotes the one predicted by the estimators trained on MNIST. The first two plots are in log scale. The label "best" reports the best one achieved by the trained estimators at each time point, whereas "specific" is the one with the random seed being $0$. ii) The second row: the two plots compare the performance of the estimators trained on MNIST with the Monte-Carlo method using Eq (2). Specifically, TrELS-SC uses trained estimators that achieve the best Spearman correlation on $D_{val}$, whereas $RD$ stands for relative difference.

