# OpenReview forum: "Faster Approximation of Probabilistic and Distributional Values via Least Squares"
_ICLR.cc/2024/Conference — ICLR 2024 poster_

### Official Review · Reviewer_3wRG · 2023-10-26

**Soundness:** 3 good
**Presentation:** 3 good
**Contribution:** 3 good
**Rating:** 8
**Confidence:** 3

**Summary:**

The paper proposes two generic estimators of the importance value of the $i$-th data point for the whole family of probabilistic values. Both estimators require $O(\frac{N}{\epsilon^2}\log\frac{N}{\delta})$ evaluations to obtain $(\epsilon,\delta)$-approximation, which is the optimal among known bounds. Further, the paper designs a framework for computing the distributional value estimator by connecting it to the obtaining least square regression. Vast experimental results show the faster convergence of the estimator over other benchmarks.

**Strengths:**

The paper extends the current results to general probabilistic values by resolving the main limitation in computing the estimator of importance value. Further, the connection between estimating distributional value and the least square estimator in (13) saves computational costs drastically. These results have a potentially significant impact on data valuation.

**Weaknesses:**

(1) One major concern I have is whether $\gamma$ (which is defined in the proof of Proposition 3 and 5) is a constant independent of $N$ for any probability values $\mathbf{p}$. It seems that setting $p_1 =1/2$ and $p_i \approx 0$ for $i>1$ yields $\gamma=O(1/N)$. As the $\gamma$ appears in the bound [(34) and (39)] for the number of evaluations, this seems critical in the first main contribution.

(2) The intuitive explanation of how the generalization to general $\mathbf{p}$ is possible should be included in the main text.

(3) Discussion on the cases when $\phi_i^{\mathcal{B},\mathbf{w}}(U)$ in Theorem 1 is close to $\phi_i^{\mathcal{D},\mathbf{w}}(U)$ would be helpful.

**Questions:**

Q1. Is $\gamma$ defined in the proof of Proposition 3 and 5 is a constant independent of $N$?
Q2. How the generalization to a general family of probability values $\mathbf{p}$ is possible?
Q3. When can $\phi_i^{\mathcal{B},\mathbf{w}}(U)$ converge to $\phi_i^{\mathcal{D},\mathbf{w}}(U)$? Could any convergences results be derived?

---

> ### Author Response · Authors · 2023-11-20
> **Response**
>
> We greatly appreciate your careful review! Below we address your concern.
>
> **Q: is $\gamma$ constant and How the generalization to probabilistic values possible?**
>
> A: Thank you for pointing out this technical issue! Indeed, $\gamma$ may not be a constant.
>
> - To simplify matters, in Appendix I we restrict to semivalues defined by a probability measure $\mu$ over $[0,1]$ to clarify our convergence analysis. Note that all probabilistic values in our experiments are semi-values. Specifically, the $\kappa(N)$ in Proposition 4 is defined by $\kappa(N) = \gamma^{-2}(N)$, which is $\Theta(1)$ if $\int_0^1 \frac{1}{t(1-t)}\mathrm{d}\mu(t) < \infty$ (e.g., Beta Shapley with $\alpha,\beta > 1$) and $O(\log^2N)$ if $\mu$ has a bounded density (e.g., Beta(4,1)). Our results provide faster estimators for the Beta Shapley values (with integers $\alpha,\beta\geq1$) that proved useful in both data valuation [2] and feature attribution [3]. More specifically, for the Beta Shapley value Beta($\alpha, \beta$), we achieve time complexity $O(\frac{N}{\epsilon^{2}}\log\frac{N}{\delta})$ if $\alpha,\beta > 1$ and $O(\frac{N}{\epsilon^{2}}\log(\frac{N}{\delta})\log^{2}N)$ if $\alpha,\beta \geq 1$.
>
> - In Appendix H, we added an interpretation for $\gamma(N)$. To sum, $\gamma(N) = \frac{\mathbb{E}\_{S}[|S|]}{N} \geq \frac1N$ where $P(S)=\cfrac{p_s+p_{s+1}}{\sum_{s=1}^{N-1}\binom{N}{s}(p_s+p_{s+1})}$ is the probability of picking $S$ by Algorithm 1. Looking into Algorithm 1, it takes each $U(S)$ to update the estimates of $|S|$ data. That means, $\gamma(N)$ is the average rate of reusing utility evaluations while running Algorithm 1. Therefore, higher average rate of reusing utility evaluations leads to more efficient ranking estimator.
>
> - As the reviewer pointed out, the worst case of the ranking estimator happens if, to be extreme, $p_{s} = 1$ if $s=1$ and $p_{s} = 0$ otherwise (corresponding to leave everthing else out). In this case, $\kappa(N) = N^2$ (equivalently, $\gamma(N)=\cfrac1N$) and our complexity bound becomes $O(N^3\log N)$, which is much worse. Interestingly, in this case, our ranking estimator in fact takes one utility evaluation to update the estimate of one data point. This indicates that our complexity bound may still have room for further improvement (at least in some cases).
>
>
>
> **Q: When can $\phi\^{\mathcal{B},\mathbf{w}}\_{i}$ converges to $\phi\^{\mathcal{D},\mathbf{w}}\_{i}$? Could any convergences results be derived?**
>
> A: If the distance between the empirical data distribution $\mathcal{B} = \frac{1}{N}\sum_{z \in D_{v}} \delta_{z}$ and the authentic data distribution $\mathcal{D}$ converges to 0 as N increases, then we think such a convergence could be possibly established. For example, Theorem 2.7 in [1] shows that $|\phi\^{\mathcal{D}\_{1},\mathbf{w}}\_{i} - \phi\^{\mathcal{D}\_{2}, \mathbf{w}}\_{i}|$ can be bounded by the Wasserstein distance of $\mathcal{D}\_{1}$ and $\mathcal{D}\_{2}$. Empirically, to have as close distance as possible, the size of $D_{v}$ (note our framework assumes $D_{v} = B$) should be large enough, and that is why we set $|D_{v}| = 10,000$ for verifying the efficacy of Algorithm 3.
>
> [1] Ghorbani, Amirata, Michael Kim, and James Zou. "A distributional framework for data valuation." International Conference on Machine Learning. PMLR, 2020.
>
> [2] Kwon, Yongchan, and James Zou. "Beta Shapley: a Unified and Noise-reduced Data Valuation Framework for Machine Learning." International Conference on AI and Statistics. 2022.
>
> [3] Kwon, Yongchan, and James Y. Zou. "WeightedSHAP: analyzing and improving Shapley based feature attributions." Advances in Neural Information Processing Systems 35 (2022): 34363-34376.

---

> > ### Comment · Reviewer_3wRG · 2023-11-22
> >
> > Thank you for the response and updates. The response is clear and my major concern is resolved. I will change my score accordingly.

---

### Official Review · Reviewer_s6uD · 2023-10-27

**Soundness:** 3 good
**Presentation:** 2 fair
**Contribution:** 2 fair
**Rating:** 6
**Confidence:** 3

**Summary:**

This paper proposes an algorithm that estimates the probabilistic value $v_i$ of each data point $i\in[n]$, i.e., $v_i:=\sum_{S\subseteq [n]\setminus \{i\}} p_{|S|}\cdot (U(S\cup \{i\})- U(S))$, where $U$ is a given utility function that maps any subset of data points to a value, and the weights $p_{k}$'s can be chosen arbitrarily by the user, as long as they satisfy $\sum_{k=0}^{n-1} p_k\cdot \binom{n-1}{k} = 1$. The well-known Shapley value is a special case of the probabilistic value. Their estimator uses $O(n\log n)$ evaluations of the utility function $U$, an improvement over the previous work [Kwon and Zou 2022] which uses $O(n^2\log n)$ evaluations.

Their estimator is based on a simple observation: because $\sum_{S\subseteq [n]} p_{|S|}\cdot U(S)=\sum_{S\subseteq [n]\setminus\{i\}} p_{|S|}\cdot U(S) + \sum_{S\subseteq [n]\setminus\{i\}} p_{|S|+1}\cdot U(S\cup\{i\})$, we have $\sum_{S\subseteq [n]\setminus\{i\}} p_{|S|}\cdot (U(S\cup \{i\})- U(S)) + \sum_{S\subseteq [n]} p_{|S|}\cdot U(S) = \sum_{S\subseteq [n]\setminus\{i\}} (p_{|S|}+p_{|S|+1})\cdot U(S\cup i)$.

Hence, their method uses random sampling to simultaneously estimate $v_i':=\sum_{S\subseteq [n]\setminus\{i\}} (p_{|S|}+p_{|S|+1})\cdot U(S\cup i)$ for all $i\in[n]$ and $\sum_{S\subseteq [n]} p_{|S|}\cdot U(S)$, and then substracts the later from the former to get the estimates of $v_i$ for all data points $i\in[n]$.

The advantage of their method is that a random sample $S\subseteq [n]$ can be used to estimate $v_i'$ for all $i\in S$ simultaneously, using just a single evaluation of the utility function -- $U(S)$. In comparison, if we use the sample $S$ to estimate the $v_i$ for all $i\in S$ directly, we would need to evaluate $v(S\setminus \{i\})$ for all $i\in S$. Thus, they get an improvement from $O(n^2\log n)$ evaluations to $O(n\log n)$ evaluations.

**Strengths:**

- The idea is simple and cute.
- The writing is fine overall.

**Weaknesses:**

- The result is technically a bit thin.
- The least squares interpretation seems a bit artificial and redundant.
- From the experiments in appendix, it is not obvious to me that the proposed estimator is any better than previous works (e.g., the well-known SHAP for estimating Shapley value).

**Questions:**

Could you elaborate the advantage of your estimator over previous works in terms of practical performance?

---

> ### Author Response · Authors · 2023-11-20
> **Response**
>
> Thank you for your feedback! Below we address your concerns.
>
> **Q: The result is technically a bit thin.**
>
> A: We agree that the idea behind Algorithm 1 is simple, but the analysis for its convergence as well as extending the idea up to Theorem 1 is not trivial. In Appendix I, we clarify conditions on which the proposed estimators enjoy the currently best complexity bound in terms of $(\epsilon,\delta)$-approximation. Empirically, in Figures 3 and 5, the proposed framework of training value estimator towards distributional values is verified to be significantly efficient and effective.
>
> **Q: The least squares interpretation seems a bit artificial and redundant.**
>
> A: We agree that the least square interpretation may not be needed if our goal is to directly approximate the valuation $\mathbf{v}$ of a *fixed* dataset. However, to cope with unseen data points through training a value estimator $\phi_\theta$, the least squares interpretation offers a clean optimization objective, which we exploited in Section 3.3: compare Eq (13) and Eq (15).
>
>
> **Q: It is not obvious that the proposed estimator is any better than previous works (e.g., the well-known SHAP for estimating Shapley value).**
>
> A: Our goal is to design *generic* estimators for all probabilistic values, instead of specializing to any particular choice, such as the Shapley value that SHAP is for. To our knowledge, before our work, the (weighted) sampling lift estimator is the only one that can apply to all probabilistic values.
>
> To our knowledege, there is no complexity bound for SHAP in terms of $(\epsilon,\delta)$-approximation. In contrast, we proved our ranking estimator enjoys the currently known best complexity $O(\cfrac{N}{\epsilon^2}\log\cfrac{N}{\delta})$. Emprically, our generic estimators are overall slightly slower than the specialized SHAP, and they are faster than the (weighted) sampling lift estimator.
>
>
> **Q: Could you elaborate the advantage of your estimator over previous works in terms of practical performance?**
>
> A: We would like to answer this question separately for i) the two proposed estimators that approximate probabilistic values, and ii) the framework of training value estimator towards distributional values.
>
> - In both data valuation [1] and feature attribution [2], the Beta Shapley values (restricted to integer $\alpha, \beta \geq 1$) tend to perform significantly better than the Shapley value. It is highly likely that the best choice often is dataset and task dependent. Thus, it is important to design faster generic estimators that work for all probabilistic values. Empirically, on Beta(4,1) (see Figures 9 and 14) and Beta(2, 2), our proposed estimators are indeed faster than the (weighted) sampling lift estimator. Note that AME applies to Beta(2,2) but not Beta(4,1).
>
>
> - Though probabilistic values were seen useful in data valuation, one big issue is that Eq. (1) cannot naturally extend to any unseen data. In contrast, the distributional value in Eq. (2) is more scalable as its definition already accounts for every possible data point. Previously, the distributional value of any unseen data point can only be approximated (and there is only one estimator for this end). Our proposed framework of training value estimators towards distributional values (Algorithm 3, which is derived from Theorem 1)  removes this hurdle, i.e., the distributional value of any unseen data point can be evaluated in one single forward-pass using a well-trained value estimator. The first two columns of Figures 3 and 5 demonstrate that the trained value estimators using Algorithm 3 learn the distributional values much better than approximation using 10,000 utility evaluations per data point. Moreover, the removal experiment, i.e., the last column of Figures 3 and 5, shows that the trained value estimators are more capable of selecting the most influential data on training, compared with the baselines.
>
>
> [1] Kwon, Yongchan, and James Zou. "Beta Shapley: a Unified and Noise-reduced Data Valuation Framework for Machine Learning." International Conference on AI and Statistics. 2022.
>
> [2] Kwon, Yongchan, and James Y. Zou. "WeightedSHAP: analyzing and improving Shapley based feature attributions." Advances in Neural Information Processing Systems 35 (2022): 34363-34376.
>
> [3] Jethani, Neil, et al. "Fastshap: Real-time shapley value estimation." International Conference on Learning Representations. 2021.

---

> ### Comment · Reviewer_s6uD · 2023-11-22
>
> Thanks for your answers!

---

### Official Review · Reviewer_XZSz · 2023-10-29

**Soundness:** 3 good
**Presentation:** 3 good
**Contribution:** 4 excellent
**Rating:** 8
**Confidence:** 2

**Summary:**

Probabilistic values, rooted in cooperative game theory, play a pivotal role in data valuation.
However, their computation poses significant challenges, especially as dataset size (N) increases.
Many current estimators either come with high computational overheads or cater only to specific instances.

Here are the main contributions of this paper:

* The authors introduce two versatile estimators, anchored in least squares regression,
  capable of efficiently approximating a broad range of probabilistic values,
  transcending the confines of earlier methods typically designed for specific values like the Shapley value.
  (Refer to Sections 3.1 and 3.2)
* Through Propositions 3 and 5, the authors establish that both novel estimators necessitate only O(N log N) utility evaluations
  to achieve a (ε,δ)-approximation, thereby aligning with the best-known computational complexities for certain cases.
* Section 3.3 unveils a pioneering approach to cast the distributional value—a more consistent alternative
  to probabilistic values—as a least squares problem.
  This breakthrough facilitates the training of machine learning models that can swiftly gauge distributional values of unseen data in a single pass.
* Validating the efficacy of their proposals, Section 4 presents experiments that
  substantiate the accelerated convergence of their estimators compared to preceding methods.
  Moreover, they successfully illustrate the potential of training models to adeptly predict distributional values for novel data points.

**Strengths:**

* The paper introduces efficient methods for both ranking and value estimation of data points.
  Through Propositions 1 and 2, they've laid out a mechanism to estimate the relative ranking underlying any probabilistic value.
  Notably, this method is computationally advantageous.
* They've established a framework, as seen in Theorem 1 and the related discussions, for training models to serve as value estimators.
  This is highly valuable, as trained models can swiftly evaluate any unseen data point
  from similar distributions in just a single forward pass, making the evaluation process much more scalable.

**Weaknesses:**

* There are computational challenges associated with training value estimators. When evaluating the distributional values, even with approximations, the computational costs are significant.
* The paper mentions that the faster convergence of certain estimators like SHAP, SHAP-paired, and the complement comes at the cost of using Θ(N^2) memory storage instead of Θ(N). However, the implications of this memory increase, especially for large-scale applications, are not discussed in depth.

**Questions:**

* While the paper presents a novel approach to data valuation using distributional Shapley values, could you shed light on the practical applications where this approach might be most beneficial?

---

> ### Author Response · Authors · 2023-11-20
> **Response (1/2)**
>
> Thank you for your efforts! Below we address your concerns.
>
> **Q: The paper mentions that the faster convergence of certain estimators like SHAP, SHAP-paired, and the complement comes at the cost of using Θ(N^2) memory storage instead of Θ(N). However, the implications of this memory increase, especially for large-scale applications, are not discussed in depth.**
>
> A: We note that this increase in memory does not apply to our methods, and we provide the following explanation for the prior methods:
> - For SHAP, it is derived from solving the problem (7). Precisely, the objective is rescaled to be an expectation, and then the objective is approximated using multiple sampled subsets. The estimate is yielded by solving the approximated objective with the efficiency constraint. The approximation includes a matrix of dimension $N\times N$, which leads to $\Theta(N^2)$ memory storage. Particularly, the inverse of this matrix is calculated at the end for aggregation, which induces $O(N^3)$ time complexity.
>
> - SHAP-paired just uses a more efficient sampling procedure to do approximation, and thus $\Theta(N^2)$ memory storage and another $O(N^3)$ time complexity are still there.
>
> - For complement estimator, they decompose Eq. (6) into $\phi_i^{Sh}(U) = \cfrac{1}{N}\sum_{k=1}^{N} \phi_{i,k}^{Sh}(U)$ where $\phi_{i,k}^{Sh}(U)=\sum_{S\colon i\in S, |S|=k}\binom{N-1}{k-1}^{-1}(U(S)-U([N]\backslash S))$. The procedure is to approximate $\phi_{i,k}^{Sh}(U)$ for every $i,k \in [N]$, and the estimate is calculated at the end by aggregating all $\{\phi_{i,k}^{Sh}(U)\}_{1\leq i,k\leq N}$, which leads to $\Theta(N^2)$ storage memory. The time complexity of the aggregation is $\Theta(N^2)$.
>
> **Q: There are computational challenges associated with training value estimators.**
>
> A: The computation time can be saved a lot by parallel computing. For MNIST, using 800 cpu cores, it takes less than two days to well approximate the ground-truth on 200 data points; less than a day to generate $\{ (S_{i}, U(S_{i})) \}$ used for training. To train a value estimator, it takes less than two days using 5 cpu cores. All in all, we believe these costs are reasonable and we hope future work will further reduce them.

---

> ### Author Response · Authors · 2023-11-20
> **Response (2/2)**
>
> **Q: Could you shed light on the practical applications where this approach might be most beneficial?**
>
> A: We would like to answer this question separately for the proposed estimators and the framework of training value estimators.
>
> - In our perspective, faster estimators for probabilistic values could potentially benefit the field of feature attribution. We have included the details in Appendix G. To be precise, there are three related works to mention:
>
>     - [1] FastSHAP was introduced as a framework for training explainers, which replaces $\mathbf{v}$ in the problem (7) (whose uniquely optimal solution $\mathbf{v}^{*}$ is the Shapley value) by trainable models. Before our work, how to cast other probabilistic values into solving optimization problems was an open question. Precisely, we answered this question by Proposition 4 (Algorithm 2 is derived from solving the problem inside this proposition). Note that AME only partially answered this question as it is restricted to a subfamily of probabilistic values which does not include, e.g., Beta(4, 1).
>
>     - [2] showed that other candidates of the Beta Shapley values (restricted to integers $\alpha,\beta \geq 1$) tend to be significantly better than the Shapley value in feature attribution. Note that [2] did not train explainers for the Beta Shapley values.
>
>     - Very recently, [3] showed that FastSHAP is equivalent to training explainers towards a biased Shapley value, and thus suggested it is better to train explainers towards the unbiased one, i.e., any estimators that converge to the Shapley value. In this view, faster estimators could potentially improve the training.
>
> - We think the framework of training value estimators towards distributional values makes data valuation methods based on marginal gains more scalable and practical.
>     - Even though probabilistic values proved handy in data valuation, their main issue is that Eq. (1) cannot be naturally extended to unseen data, which makes it intractable to train value estimators for probabilistic values.
>     - The value assinged to each data point indicates its quality of some type (e.g., the extent of being an outlier), and the type of quality that our framework determines is through the aggregation of marginal gains $U(S\cup i)-U(S)$. In other words, the choice of utility function $U$ is critical for the final performance of the considered task. Therefore, there are possibilities considering that the utility function $U$ in our framework only serves as a module, such as detecting outlies, the most influential data, etc.\.
>
> [1] Jethani, Neil, et al. "Fastshap: Real-time shapley value estimation." International Conference on Learning Representations. 2021.
>
> [2] Kwon, Yongchan, and James Y. Zou. "WeightedSHAP: analyzing and improving Shapley based feature attributions." Advances in Neural Information Processing Systems 35 (2022): 34363-34376.
>
> [3] Zhang, Borui, et al. "Exploring Unified Perspective For Fast Shapley Value Estimation." arXiv preprint arXiv:2311.01010 (2023).

---

> > ### Comment · Reviewer_XZSz · 2023-11-22
> >
> > Thank you for the response and updates. I left the score unchanged.

---

### Official Review · Reviewer_wdkt · 2023-11-01

**Soundness:** 3 good
**Presentation:** 2 fair
**Contribution:** 3 good
**Rating:** 8
**Confidence:** 3

**Summary:**

The paper proposed two generic probabilistic values estimators (one for ranking and the other for exact probabilistic values) that achieves $O(\frac{N}{\epsilon^2} \log \frac{N}{\delta})$ utility evaluations, improving upon previously known generic estimators that require $O(\frac{N^2}{\epsilon^2} \log \frac{N}{\delta})$. The authors prove theoretically convergence rates of their estimators, and present their performance with numerical experiments.

**Strengths:**

The result is interesting and novel to me, though I am not an expert in this area. The estimators proposed is general enough, and comparisons with previous literature is thorough. The paper provides both good theoretical and numerical evidence.

**Weaknesses:**

1. Lack of direct comparisons to previous estimators specific to special cases achieving the same rate. I would like to see more discussions on SHAP, MSR and AME estimators. Why they cannot be generalized? What are their main ideas comparing to this paper, what is the fundamental differences?
2. It may be that I'm not familiar with the field, but the presentation does not properly introduce the background. It would be nice if the authors can provide a few more demonstrations in the introduction, i.e. examples for Shapley and Banzhaf. I did not see proper explanations on those probabilistic values while reading the paper.
3. The numerical plots could look better in log-scale, for now it's hard for me to distinguish between lines when they converge.
4. Lack of expanding on the theoretical results. Some explanations on the intuition on why an $\Theta(N/\epsilon^2)$ convergence rate is expected and what makes it different from previous estimators would be nicer.

**Questions:**

In conjunction with 1-4 in the weakness part.

---

> ### Author Response · Authors · 2023-11-20
> **Response (1/2)**
>
> Thank you for your helpful feedback. We have re-plotted all the convergence results for comparing estimators in log-scale in the revision. Below we address your comments.
>
> **Q: Lack of direct comparisons to previous estimators specific to special cases achieving the same rate. I would like to see more discussions on SHAP, MSR and AME estimators. Why they cannot be generalized? What are their main ideas comparing to this paper, what is the fundamental differences?**
>
> A: We detail the difference below.
>
> - MSR: It does sampling based on Eq. (4). Precisely, it repeats the procedure of sampling a subset $S$ by including each data point with probability $0.5$. With $T$ sampled subsets $\lbrace{S_j\rbrace}\_{1\leq j\leq T}$, the estimate for the $i$-the data point is
> $\cfrac{1}{|\mathcal{S}\_{\ni i}|}\sum_{S\in \mathcal{S}\_{\ni i}}U(S)-\cfrac{1}{|S_{\not\ni i}|}\sum_{S\in S_{\not\ni i}}U(S)$ where $\mathcal{S}\_{\ni i}=\lbrace{S_j \mid i \in S_j\rbrace}$
> and $\mathcal{S}_{\not\ni i} = \lbrace{S_j \mid i\not\in S_j\rbrace}$. In terms of $(\epsilon,\delta)$-approximation, its time complexity is $O(\cfrac{N}{\epsilon^2}\log\cfrac{N}{\delta})$, ours enjoy the same time complexity as proved in Corollary 1.
>     - Difference: MSR only applies to probabilistic values that satisfy $\frac{p_{s+1}}{p_{s}} = C$ for every $1\leq s< N$ (see Appendix C.2 in [1]) while ours are for all probabilistic values.
>     - Why it cannot be generalized? Not every probabilistic vlaue corresponds to a $P_{MSR}$ such that Eq. (1) can be rewritten as Eq. (4).
>
>
> - SHAP: It is derived from solving the estimate of the problem (7). Precisely, the objective there is scaled into an expectation (which does not modify its uniquely optimal solution, the Shapley value). Then, subsets are sampled according to the scaled weights to approximate the objective, after which the estimate is calculated by solving the estimated objective with the efficiency constraint.
>     - Difference: i) To our knowlege, there is no  complexity bound for SHAP in terms of $(\epsilon,\delta)$-approximation, while our ranking estimator has complexity $O(\cfrac{N}{\epsilon^2}\log\cfrac{N}{\delta})$. ii) SHAP is to approximate the Shapley value while our methods work with any probabilistic value.
>     - Generalizations: Ours may be seen as generalizations of SHAP as they are derived from solving the problems (10) and (11). Precisely, the ranking estimator is to approximate $\mathbf{b}$ (up to some scalar) in $\mathbf{A}^{-1}\mathbf{b}$  which represents the uniquely optimal solution to the problem (10).
>
> - AME: It is derived from solving the estimate of the problem (8). Its objective is already an expectation. So, it uses multiple sampled subsets to approximate the objective, with which the approximate optimal solution is obtained. This solution represents the estimate of the underlying probabilistic values.
>     - Differnce: i) For every probability measure $P$ over $[0,1]$, the definition $p_s=\int_0^1 t^{s-1}(1-t)^{N-s}\mathrm{d}P(t)$ for $\mathbf{p}$ in Eq. (1) yields a probabilistic value, which is the uniquely optimal solution to the problem (8) while using $P$ to sample $\mathbf{X}$ and $Y$; if $M_{P} := \mathbb{E}_{t\sim P}[\cfrac{1}{t(1-t)}] = \infty$, the correponding probabilistic value cannot be approximated by AME. Examples in our experiments include the Shapley value and Beta(4,1). ii) By assuming that $P$ has support $[a,b]$ where $a>0$ and $b<1$, they proved AME has complexity bound $O(N\log N)$. Under this assumption, ours achieve the same bound. However, the Beta Shapley value Beta($\alpha, \beta$), with integers $\alpha, \beta\geq 1$, does not verify this assumption, but we can still prove that ours achieve $O(N\log N)$ if $\alpha,\beta>1$ and $O(N\log^3 N)$ if $\alpha,\beta \geq 1$.
>     - Why it cannot be generalized? For each sampled subset $S$, $\widetilde{\mathbf{X}} \in \mathbb{R}^{N}$ is defined to be $\widetilde{X}_i = \cfrac{1}{t}$ if $i\in S$ and $\cfrac{-1}{1-t}$ otherwise, whereas $Y=U(S)$. Then, the optimal solution to the problem (10) is $M_P \mathbb{E}[\widetilde{\mathbf{X}}\widetilde{\mathbf{X}}^{\top}]^{-1}\mathbb{E}[\widetilde{\mathbf{X}}Y]$. Specifically, the diagonal entries of $\mathbb{E}[\widetilde{\mathbf{X}}\widetilde{\mathbf{X}}^{T}]$ are all $M_P$, which will blow up if $M_P=\infty$.

---

> ### Author Response · Authors · 2023-11-20
> **Response (2/2)**
>
> **Q: It would be nice if the authors can provide a few more demonstrations in the introduction.**
>
> A: We added Appendix F on this topic. For each probabilistic value Eq. (1), the only constraint is that $\mathbf{p}$ is non-negative and $\sum_{s=1}^{N}\binom{N-1}{s-1}p_{s} = 1$. But for semi-values (a subfamily of probabilistic values), we have $p_{s} = \int_{0}^{1} t^{s-1}(1-t)^{N-s}\mathrm{d}\mu(t)$ where $\mu$ is a probability measure on $[0,1]$. Conversely, each probability measure leads to a semi-value through the same formula. To our knowledge, all  probabilistic values in the previous references are all semi-values. For example, $\mu$ uniform produces the Shapley value, i.e., $p_{s} = \frac{(s-1)!(N-s)!}{N!}$; $\mu = \delta_{0.5}$ (Dirac delta distribution) generates the Banzhaf value, i.e., $p_{s} = \frac{1}{2^{N-1}}$; beta distribution with pdf $\propto t^{\beta-1}(1-t)^{\alpha-1}$ leads to Beta$(\alpha,\beta)$ (notation for Beta Shapley values), i.e., $p_{s} = \frac{\Gamma(\alpha+\beta)}{\Gamma(\alpha)\Gamma(\beta)}\cdot\frac{\Gamma(\beta + s - 1)\Gamma(\alpha+N-s)}{\Gamma(\alpha+\beta+N-1)}$. In [2-3], the employed Beta Shapley values restricted to integers $\alpha, \beta \geq 1$.
>
>
> **Q: Lack of expanding on the theoretical results. Some explanations on the intuition on why an $\Theta(\frac{N}{\epsilon})$ convergence rate is expected and what makes it different from previous estimators would be nicer.**
>
> A: We have added Appendix I to clarify our convergence analysis. We mainly compare our proposed estimators to the (weighted) sampling lift estimator as (to our knowledge) it is the only *generic* estimator before our work. To compare,
> ours use one utility evaluation $U(S)$ to update the estimates of $|S|$ (in expectation $N \gamma(N)$, see Appendix H) data, while the (weighted) sampling lift estimator takes two utility evaluations to update only *one* data point. In this view, ours are more efficient in reusing each utility evaluation, which explains  its faster convergence. Specifically, for Beta($\alpha,\beta$), ours achieve $O(\cfrac{N}{\epsilon^2}\log\cfrac{N}{\delta})$ if $\alpha,\beta>1$ and $O(\cfrac{N}{\epsilon^2}\log(\cfrac{N}{\delta})\log^2 N)$ if $\alpha,\beta\geq 1$. Previously, to our knowledge, only the (weighted) sampling lift （which requires $O(\cfrac{N^2}{\epsilon^2}\log\cfrac{N}{\delta})$ instead） and AME (which only works with $\alpha,\beta>1$) can be used for the Beta Shapley. Though it was proved that AME enjoys $O(\cfrac{N}{\epsilon^2}\log\cfrac{N}{\delta})$ under certain assumption, the Beta Shapley dose not verify this assumption.
>
>
> [1] Wang, Jiachen T., and Ruoxi Jia. "Data banzhaf: A robust data valuation framework for machine learning." International Conference on Artificial Intelligence and Statistics. PMLR, 2023.
>
> [2] Kwon, Yongchan, and James Zou. "Beta Shapley: a Unified and Noise-reduced Data Valuation Framework for Machine Learning." International Conference on AI and Statistics. 2022.
>
> [3] Kwon, Yongchan, and James Y. Zou. "WeightedSHAP: analyzing and improving Shapley based feature attributions." Advances in Neural Information Processing Systems 35 (2022): 34363-34376.

---

> ### Comment · Reviewer_wdkt · 2023-11-22
>
> Thank you for the update and it addresses my concerns. I'm raising my score accordingly.

---

### Author Response · Authors · 2023-11-20
**Global Comments**

We thank all reviewers for their efforts and constructive feedback! Here we summarize the updates in the revision. All changes in the main paper and appendix were marked in blue.

- Figure 1, Figures 6 - 14 that compare the convergence of estimators have been re-plotted in log-scale.

- Appendix I is to clarify our convergence analysis, i.e., Propositions 4 and 5. As pointed out by Reviewer 3wRG, our  complexity bounds contain terms $\kappa(N)$ and $\tau(N)$ that may depend on $N$. In Appendix I, we restrict to semivalues defined by a probability measure $\mu$ on $[0,1]$ and we show that $\kappa(N)=\tau(N)=\Theta(1)$ if $\int_0^1 \frac{1}{t(1-t)}\mathrm{d}\mu(t) < \infty$ (e.g., Beta-Shapley for $\alpha,\beta > 1$) while $\kappa(N)=\tau(N)=O(\log^2N)$ if $\mu$ has a bounded density (e.g., Beta(4,1)). We note that all probabilistic values employed in our experiments and in the literature are semivalues. Moreover,  faster estimators (except AME) than the straightforward sampling lift method are not designed for the Beta Shapley values, a gap that we fill with the best known complexity bound.

- Appendix H is to provide an intuitive interpretation on $\kappa(N)$ that appears in Proposition 4. To conclude, it is the inverse square of the average rate of reusing utility evalutions while running Algorithm 1.

- Appendix J is to add more introduction to probabilistic values used in practice.

- Appendix G is to comment on why faster estimators for probabilistic values (instead of for specific cases) could benefit the field of feature attribution.

---

### Meta-Review · Area_Chair_hz4o · 2023-12-05

**Metareview:**

This paper proposes generic probabilistic values estimators that incur less utility evaluations to achieve the same approximation quality as existing ones.

STRENGTH

The technical contributions are novel, interesting, and non-trivial.


WEAKNESS

(1) There are some clarification issues (e.g., explanations for the theoretical results, direct comparisons with existing estimators, discussion of practical applications, gamma being a constant, etc) that have been addressed well by the authors in the rebuttal.


(2) I would also like the authors to discuss in their revised paper whether and how well the corresponding axioms can be satisfied after applying their proposed approximations. See, for example, the following missing reference:

Probably Approximate Shapley Fairness with Applications in Machine Learning. AAAI 2023.

The authors should also comment on the difference in approximation quality of their proposed estimator from that in the above reference, and discuss any implications.

**Justification For Why Not Higher Score:**

Though the results are new, the practical significance may be limited since the utility evaluations remain computationally costly, especially when scaling up to large data.

**Justification For Why Not Lower Score:**

The proposed estimators are non-trivial improvements over the existing ones.

---

### Decision · Program_Chairs · 2024-01-16

Accept (poster)